# Efficient Rotation Invariance in Deep Neural Networks through Artificial Mental Rotation

## Abstract

Humans and animals recognize objects irrespective of the beholder's point of view, which may drastically change their appearance. Artificial pattern recognizers strive to also achieve this, e.g., through translational invariance in convolutional neural networks (CNNs). However, CNNs and vision transformers (ViTs) both perform poorly on rotated inputs. Here we present AMR (artificial mental rotation), a method for dealing with in-plane rotations focusing on large datasets and architectural flexibility, our simple AMR implementation works with all common CNN and ViT architectures. We test it on randomly rotated versions of ImageNet, Stanford Cars, and Oxford Pet. With a top-1 error (averaged across datasets and architectures) of 0.743, AMR outperforms rotational data augmentation (average top-1 error of 0.626) by 19%. We also easily transfer a trained AMR module to a downstream task to improve the performance of a pre-trained semantic segmentation model on rotated CoCo from 32.7 to 55.2 IoU.

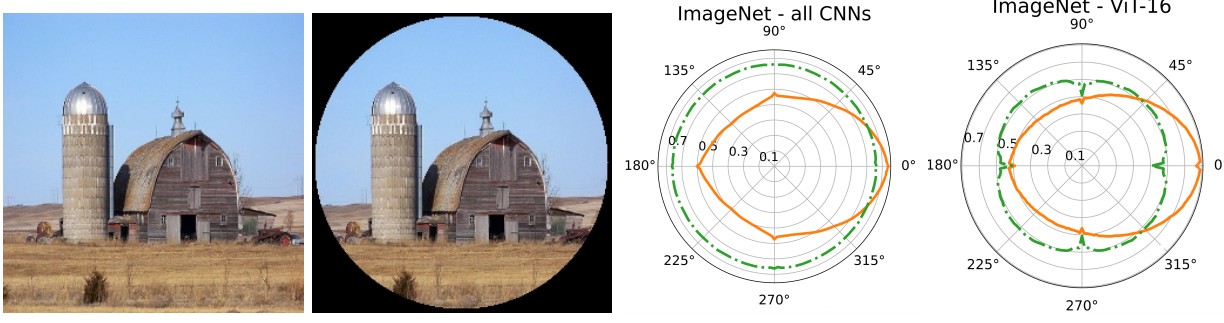

Figure 1: An example image from ImageNet (far left) and a version of the same image with its corners masked, to allow for non-obvious and reversible rotations (center left). Polar plots of ImageNet top-1 accuracies by angle averaged across CNN eight architectures (center right) and for VIT-16 (far right). The performances in solid orange are for models with standard training, the dash-dotted green lines represent training with rotational data augmentation. For training details see Chapter 3.

## 1 Introduction

Natural vision systems in humans and animals are able to recognize objects irrespective of transformations such as rotations and translations as well as the observer's point of view, all of which can have a tremendous impact on the appearance of said object. This is a highly desirable property for all vision systems, especially if they are to be deployed in real-world settings that are characterized by significant scale and visual complexity (Stadelmann et al., 2018; 2019). CNNs (Fukushima et al., 1980; 1983; Waibel et al., 1989; Zhang et al., 1988) inherently integrate translational invariance into their design. Vision transformers Vaswani et al. (2017); Dosovitskiy et al. (2021) (a.k.a. neural fast weight programmers (Schmidhuber, 1992; Schlag et al., 2021)) exhibit a level of translational robustness however they are not fully translational invariant (Rojas-Gomez

et al., 2024). For rotations, this is not the case and both methods perform very poorly when facing inputs at an unusual angle (see Figure 1 and (Engstrom et al., 2019; Xu et al., 2023)). This can be exploited for adversarial attacks (Engstrom et al., 2019) and thus can cause serious issues in applications where rotated inputs are common.

A widely used approach is based on input data augmentation. The data is rotated at training time such that the model can learn all appearances of an object. This yields good results and can scale to any problem size. It is, however, still an inefficient method reducing the sample efficiency and consequently resulting in lower final performance with equal training time (see differences at angle zero in Figure 1).

A straightforward approach to achieving rotation invariance focuses on building architectures that incorporate rotational invariance or sometimes equivariance directly into the neural network design (Cohen & Welling, 2016; Marcos et al., 2017; Dieleman et al., 2015; Cohen & Welling, 2022; Weiler & Cesa, 2019; Laptev et al., 2016; Worrall et al., 2017; Kaba et al., 2023; Mondal et al., 2023). However, the largest gains in model performance in the last years have been realized through systematic scaling up of model and training data size as well as training time (Tan & Le, 2019; Zhai et al., 2022). This trend led to growing model architectures and culminated in the inception of foundation models (Bommasani et al., 2021; Kirillov et al., 2023). These large and ever-evolving architectures are generally not rotation invariant (Mondal et al., 2023). We focus here on practical applicability and performance so we aim to leverage the power of these models directly, without change. To achieve this a decoupled method facilitating the addition of rotation invariance is needed to solve the problem of degraded performance in the presence of in-plane rotations in an architecture-independent fashion. This demands methodological simplicity with low development overhead and fast execution w.r.t. runtime on top of excellent performance on various downstream vision tasks (e.g. classification or segmentation).

It is a long-standing conjecture in neuro-psychology that when humans try to identify an object, they mentally simulate rotations of that object to match it to an internal representation (i.e. they perform mental rotation). Shepard & Metzler (1971) were the first to formally study this phenomenon. They were able to show that the time human subjects need to determine whether pairs of 3D figures have the same shape grows linearly with the angle of rotation between the two objects. This strongly suggests that humans perform mental rotation; otherwise, the re-identification task would be completed in constant time across angles. This concept of mental rotation has been of inspiration to several computer vision methods Ding & Taylor (2014); Boominathan et al. (2016); Feng et al. (2019); Fang et al. (2020); Kaba et al. (2023); Mondal et al. (2023).

In this paper, we introduce the Artificial Mental Rotation (AMR) method that separates the finding of the angle of rotation of a given input from subsequent rotating it back to its canonical appearance before further processing, thus performing an artificial version of mental rotation. The problem of rotation estimation has been considered hard in the literature before (Boominathan et al., 2016). However, it has the advantages that the angles can be found in a one-shot fashion and the underlying method of visual recognition itself does not have to be hardened against rotations, therefore all models (even trained ones) can be used in conjunction with an AMR module.

In short, our core contributions are: (a) We introduce a simple approach and corresponding self-supervised training method, AMR, for invariant processing of rotated images, (b) we present a simple neural network architecture that implements AMR and can be paired with all common CNNs and ViTs without alteration of the trained base model, (c) we extensively test the real-world merits of AMR on rotated versions of ImageNet, Stanford Cars, and Oxford Pet, and conclude that it significantly outperforms rotational data augmentation and generally shows excellent performance in practically relevant tasks, (d) we present AMR results on MNIST showing it performs competitively to existing methods, (e) we confirm the viability of AMR in a scenario where only portions of the test data are rotated, (f) we present comprehensive ablation studies proofing that our trained AMR modules work in practice on synthesized as well as physically rotated data, and (g) we show the easy transferability of a trained AMR module to another downstream vision task (in this case semantic segmentation), significantly increasing the performance of an existing model on rotated data.

## 2 Related Work

An important early work is Spatiatial Transformer Networks (Jaderberg et al., 2015) which extends CNNs with the ability to learn spatial transformations (including rotations) for its feature maps. This architecture has since been tailored to specific equivariances (Esteves et al., 2018; Tai et al., 2019).

There are ongoing efforts to incorporate rotation invariance (or in some cases equivariance) directly into the architectures of deep neural networks, especially for CNNs. Dieleman et al. (2015) introduced a rotation invariant CNN system for galaxy morphology prediction that uses multiple rotated and cropped snippets of the same image as input. Cohen & Welling (2016) presented G-CNNs which are equivariant to a larger number of symmetries such as reflections or rotations. This is achieved by lifting the network's features to a desired symmetry group. The later work on steerable CNNs (Cohen & Welling, 2022; Weiler & Cesa, 2019) extended this work. Romero & Cordonnier (2021) presented group equivariant vision transformers by extending the symmetry group lifting concept to self-attention. Worrall et al. (2017) introduced H-Nets which replace regular CNN filters using circular harmonics. Marcos et al. (2017) have proposed to rotate the filters of a CNN and then apply spatial and orientation pooling to reduce and merge the resulting features. Laptev et al. (2016) introduced a TI-pooling, which allows to pool the CNN outputs for an arbitrary number of different angled versions of the same input to create an equivariant feature. These methods all entangle model architecture and equivariance properties.

Data augmentation (Baird, 1992) is very widely used to improve the robustness and generalizability of vision models (Simard et al., 2003). It can even be used to harden the model against adversarial attacks (Shafahi et al., 2019). Data augmentation has also been shown to be very effective for rotated inputs (Quiroga et al., 2020). Later work aims to improve the sample efficiency of data augmentation by directly learning object-specific invariances or transformation inference functions to inform the augmentation process (Miao et al., 2023; Immer et al., 2022; Allingham et al., 2024). While improving rotational stability do data augmentation approaches not tackle the issue on a fundamental level.

There have been previous attempts to leverage the concept of mental rotation for computer vision. Ding & Taylor (2014) trained a factored gated restricted Boltzmann machine to actively transform pairs of examples to be maximally similar in a feature space. Boominathan et al. (2016) train a shallow neural network to classify if an image is upright. They combine this with a Bayesian optimizer to find upright images and use this setup to improve image retrieval robustness. In the space of 3D vision, a mental rotation-based approach achieved state-of-the-art performance for rotated point cloud classification (Fang et al., 2020). In the representation learning community rotation prediction using CNNs has been leveraged as an additional, self-supervised, learning signal to train better representations (Feng et al., 2019). Kaba et al. (2023) achieve invariance via learned canonicalization functions. In a follow-up work Mondal et al. (2023) adapt canonicalization functions to large pretrained models. While being unique, both of these works highly relate to this contribution, this relation is discussed in more detail in Section 7.

## 3 Artificial Mental Rotation

Our AMR approach requires three components. First, a base model (BM) is required, for which any common CNN or ViT (Dosovitskiy et al., 2021) architecture can be used. There is no need to modify the BM in any way, hence the BM can generally be sourced in a fully (pre-)trained form. However, we do copy features out of the BM at various stages, in cases where this is not possible e.g. when using a BM hidden behind an API the AMR module has to be designed as a stand-alone network (this would be equivalent to Stem in Chapter A). Additionally, it requires a rotation algorithm designed for images; here we use the method available in OpenCV (Bradski, 2000). The last necessary component is the AMR module itself, presented in this section. Due to their differing designs, CNNs and ViTs use slightly varying AMR modules.

**AMR training** While training the AMR module, the BM is frozen such that its classification performance is not disturbed. For the training, we use datasets, like ImageNet, where the objects are typically shown in an upright position. Under this constraint, we can employ self-supervised training by randomly rotating the input images and asking the AMR module to recover the angle we previously applied.

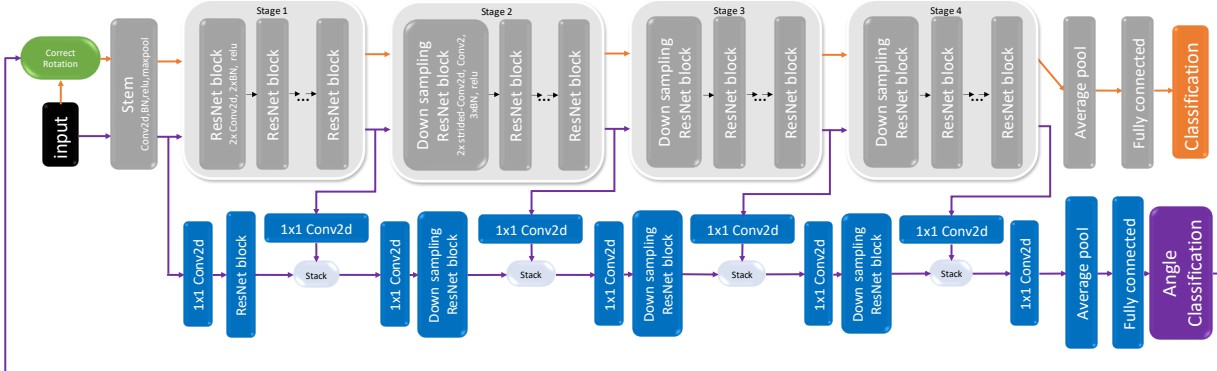

Figure 2: Architecture of our artificial mental rotation module for CNNs. The base CNN, in this case a ResNet, is shown in grey. The components of the AMR module are shown in blue. The information flow in stage 1 (angle classification) is purple while the information flow in stage 3 (image classification) is shown in orange.

**AMR inference**   AMR inference is performed in a three-step process: (1) The input's angle is classified by running it through the BM and the AMR module. (2) The input is rotated by the negative amount of the angle determined in step one. (3) The rotation-corrected input is processed by the BM.

Step (3) is identical to AMR-free inference since the BM is frozen during AMR training and there is no information flow through the AMR module during this step. Therefore AMR could also be framed as a preprocessing method by reducing it to steps (1) and (2).

## 3.1   AMR module for CNNs

Our AMR module is designed as an add-on to a given BM (see Figure 2), so it can repurpose the features computed by the BM and only requires a small number of additional weights. Features are copied into the AMR module at five different BM stages. In the case of ResNe(X)ts (He et al., 2016; Xie et al., 2017) (a.k.a. Highway Nets with open gates Srivastava et al. (2015)) this happens directly after the stem and after each of the four ResNe(X)t stages. For EfficientNets (Tan & Le, 2019) we use the end of stages 2, 4, 6, and 8 as extraction points. When the copied features enter the AMR module they are first processed by a single $1 \times 1$ 2D convolution to compress the feature depth. For all but the first AMR module stages, these features are then stacked with the output of the previous AMR module stage followed by another $1 \times 1$ 2D convolution to half the feature depth of the stack. Only then the data is processed by a single ResNet block. After the last stage, we employ average pooling and a single fully connected layer with 360 outputs to create the angle prediction.

## 3.2   AMR module for ViTs

The AMR module for ViTs functions very similarly to the one for CNNs. We again extract features at five different locations. For ViT-16-b these are after encoder blocks 1, 4, 7, and 12. Since there is no spatial downsampling in ViTs there is no advantage in processing the extracted features in stages. We, therefore, stack them all at once followed by a single $1 \times 1$ 2D convolution. This stack is then processed by four ViT encoder modules. Lastly, we extract the same classification token that was used in the BM and apply a fully connected layer for the angle classification.

## 3.3   Motivation for add-on design

We opted to design our AMR module as an add-on to existing base networks because we conjecture that the features that have been trained for classification will also be at least partly useful for angle detection

Table 1: ImageNet top-1 accuracies. Upright testing (up) of the upright trained base model is assumed to be the performance ceiling (% ceil). Average (by angle) rotated accuracies (rot) are given for the upright and rotated trained base models as for AMR 33 epochs and AMR 5 epochs.

| Testing | Upright Training | | | Rotated Training | | AMR 33 | | AMR 5 | |
|---|---|---|---|---|---|---|---|---|---|
| | up | rot | % ceil | rot | % ceil | rot | % ceil | rot | % ceil |
| ResNet-18 | 0.695 | 0.433 | 62 | 0.598 | 86 | **0.676** | **97** | 0.666 | 96 |
| ResNet-50 | 0.768 | 0.537 | 70 | 0.673 | 88 | **0.755** | **98** | 0.746 | 97 |
| ResNet-152 | 0.779 | 0.552 | 71 | 0.730 | 94 | **0.767** | **98** | 0.760 | **98** |
| EfficientNet-b0 | 0.680 | 0.454 | 67 | 0.611 | 90 | **0.666** | **98** | 0.656 | 96 |
| EfficientNet-b2 | 0.692 | 0.467 | 67 | 0.612 | 88 | **0.678** | **98** | 0.669 | 97 |
| EfficientNet-b4 | 0.710 | 0.485 | 68 | 0.618 | 87 | **0.696** | **98** | 0.689 | 97 |
| ResNext-50-32x4d | 0.773 | 0.551 | 71 | 0.686 | 89 | **0.761** | **98** | 0.754 | 98 |
| ResNext-101-32x8d | 0.785 | 0.571 | 73 | 0.728 | 93 | **0.772** | **98** | 0.766 | **98** |
| ViT-16b | 0.691 | 0.459 | 66 | 0.503 | 73 | **0.669** | **97** | 0.664 | 96 |
| Average | 0.730 | 0.501 | 69 | 0.640 | 87 | **0.716** | **98** | 0.708 | 97 |

and the AMR module can profit from the training resources that have already been invested into the base network. We confirm this conjecture with an ablation study (see Section A in the Appendix). This design choice therefore allows for an AMR module that consists of very few layers on its own. Thus it can be trained very quickly and only adds a constant overhead of roughly 5 Million parameter resulting in 0.905 GFlops.

## 4 Experiments

We aim to showcase the merits of AMR on natural images. Therefore, we test it on ImageNet (ILSVRC 2012) (Russakovsky et al., 2015) and verify our results on Stanford Cars (Krause et al., 2013) and Oxford Pet (Parkhi et al., 2012), by employing rotated versions of the mentioned datasets. To ensure that artificial rotations are not obvious, we mask out the corners of all images such that a centred circle remains (see Figure 1 and the next Section for an ablation study ensuring the artificial rotations are not carrying any unwanted information). For a fair comparison between upright training (without data augmentation) and training with random rotations as input data augmentation (rotated training), we train all of our base models from scratch with this masking applied. For all of our training runs, we use image normalization based on dataset statistics. No further data augmentation is applied to keep the experiments as simple as possible (except, of course, input rotation for the rotated training models). To obtain representative results we replicate our experiments on a variety of base models. We use three different ResNets, three EfficentNets, and two ResNeXts for a total of eight CNN architectures. On ImageNet we also employ a vision transformer in the form of ViT-16b, which is unsuited for the other smaller datasets. For each upright trained base model, we train two AMR modules: One is trained for one-third of the base model's training time (in epochs) and the other one for one-twentieth.

**Training details** For all base models, we use the implementations from the torchvision (maintainers & contributors, 2016) Python package without any modifications. To enable optimal training speed our code is based on the ffcv library (Leclerc et al., 2022). All training details and links to code and trained model weights can be found in the Appendix.

**Testing** We first evaluate the upright base models on upright data. We use these performances as the ceiling of what can be achieved on rotated data. Then we test the upright and rotated base models as well as the AMR-enhanced models for rotated performance by rotating the test set two degrees at a time and running a full evaluation for each angle. We present the resulting data visually in polar plots (see Figure 3 and Figure 4) as well as in table form (see Table 1 and Table 2) by averaging across angles.

**ImageNet** We train all of our base CNNs for 100 epochs on ImageNet, and the vision transformer is trained for 300 epochs, in accordance with the training recipes for the torchvision base models. We then train two AMR modules in conjunction with each upright trained base model, one for 33 epochs and the other for 5.

Table 2: Stanford Cars and Oxford Pet top-1 accuracies averaged across all architectures, columns are analogous to the Table 1 shown above for ImageNet.

|  | Upright Training | | | Rotated Training | | AMR 300 | | AMR 50 | |
| --- | --- | --- | --- | --- | --- | --- | --- | --- | --- |
| Testing | up | rot | % ceil | rot | % ceil | rot | % ceil | rot | % ceil |
| Stanford Cars | 0.867 | 0.165 | 19 | 0.618 | 71 | **0.796** | **92** | 0.746 | 86 |
| Oxford Pet | 0.741 | 0.483 | 65 | 0.603 | 81 | **0.712** | **96** | 0.670 | 90 |

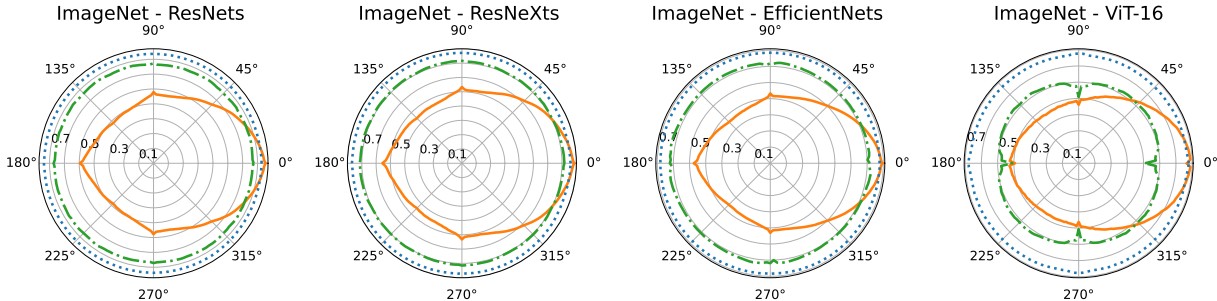

Figure 3: Polar plots of ImageNet top-1 accuracies by image rotation angle, averaged across architectures (except ViT). The performances of the upright base models are shown in solid orange, the rotated training base models are shown in dash-dotted green, and AMR performance (averaged across both epoch regimes) is shown in dotted blue lines.

Table 1 contains the top-1 accuracies on rotated data for all models. Additionally, the ceiling accuracy is also reported (upright data with upright trained model). As suspected, there is a steep drop in accuracy between upright and rotated testing for the upright-trained models, both for the CNNs as well as the ViT. On average only 69 percent of the ceiling performance (% ceil) is retained. The models which have been trained with random rotations fare much better, they achieve 87% ceil. It is noteworthy that the ViT only rises from 66 to 73 of the ceiling performance. This makes sense since ViTs tend to be less sample efficient compared to CNNs and therefore suffer more from the increased problem complexity caused by the random rotations. AMR-33 achieves 98% ceil, significantly outperforming rotated training. AMR-5 is slightly worse with 97% ceil, but it shows that it is possible to obtain an AMR module that is very useful with minimal training resources. Figure 3 contains polar plots that show the ImageNet top-1 accuracies of the different architecture families by angle. The solid orange lines show the accuracies of the upright-trained base models. We observe that the accuracies have their highest points at zero degrees rotation and then symmetrically drop off with increasing angle, reaching their lowest points at 135 and 225 degrees. We further observe that rotated training (green dash-dotted line) and AMR (blue dotted line) both achieve rotational invariance and exhibit performances that are independent of test time angles. Corresponding to the reported results in Table 1, AMR performance is consistently better than rotated training.

**Stanford Cars and Oxford Pet** Due to Stanford Cars and Oxford Pet being smaller datasets we forgo ViTs and train the CNN models for more epochs on Stanford Cars and Oxford Pet. On Stanford Cars we train the base models for 1000 epochs, and the corresponding AMR modules are trained for 300 and 50 epochs, respectively. On Oxford Pet, we train the base models for 3000 epochs and the AMRs for 1000 and 150 epochs. Table 2 shows the top-1 accuracies averaged across architectures and averaged across angles where appropriate (for full table see Table 5 in the Appendix) and Figure 4 polar plots, analogous to the ones for ImageNet in the above paragraph. Our core findings are replicated on both datasets: Rotating the images reduces all the models' performances and AMR remains the more powerful way of addressing rotations on these datasets. On Stanford Cars, the performance loss caused by rotations on the upright trained model is much more severe (19% ceil) compared to ImageNet (69% ceil), with the models failing almost completely when facing rotations larger than 20 degrees (see left column of Figure 4). This makes sense intuitively since cars are almost always upright in pictures with minimal variation, thus the models experience almost no variation during training. This is further supported by the observation that Oxford Pet, which is also a

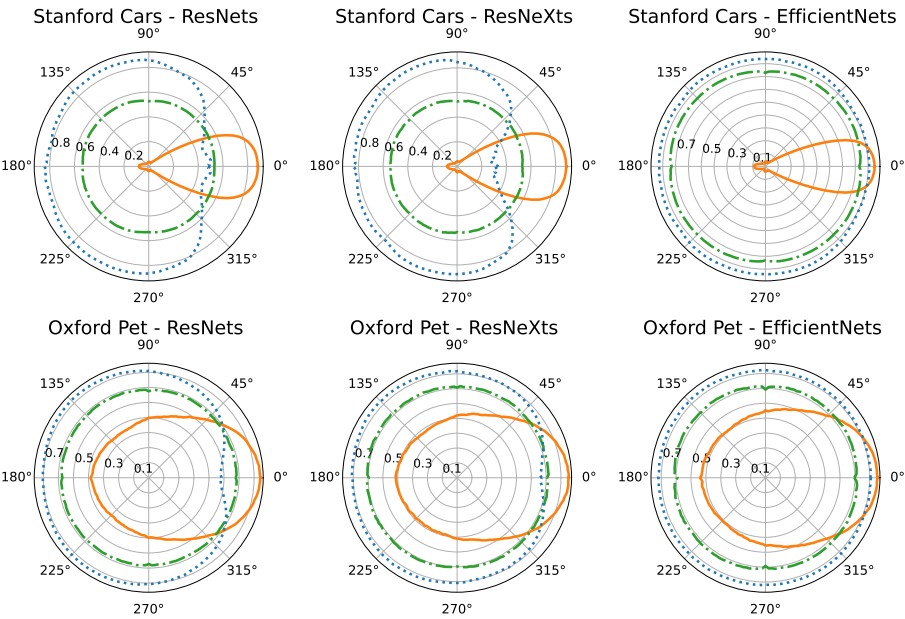

Figure 4: Polar plots of Stanford Cars (left column) and Oxford Pet (right column) top-1 accuracies analogous to the ones shown above for ImageNet (see Figure 3).

small dataset but contains animals that are naturally less static compared to cars, exhibits a milder drop off (65% ceil). We further observe that on Stanford Cars and to a lower extent on Oxford Pet, EfficientNets perform much better than ResNe(X)ts on rotated data, both with rotated training and AMR, while all architectures perform roughly equally well on upright data. We conjecture this is because EfficientNets have been designed to be sample efficient. This could allow them to train filters useful for a wide variety of tasks (such as AMR) even on a small dataset and a relatively short training time. However, an unexpected result is that AMR paired with ResNe(x)t models showed a decline in performance when approaching 0 degrees, while EfficientNets do not suffer from this effect. In the Appendix, we investigate this phenomenon further.

**Comparison with existing rotation equivariant methods on MNIST**    The focus of AMR is modern, large architecture and correspondingly large datasets. The current literature for rotation equivariant methods is focused on rotated MNIST as the benchmark dataset of choice for most of these methods. To put our work into perspective with these related works we present the performances of ResNet-18 (He et al., 2016), ResNet-18 + rotated training and ResNet18+AMR on MNIST (see Table 3). The ceiling performance of ResNet18 on upright MNIST is almost one, which is to be expected. Similar to the larger datasets above is the performance of AMR superior to rotated training, however only by a small margin. This makes sense intuitively since for such a simple dataset the reduced sample efficiency of rotated training plays a small role. Most importantly, the performance of ResNet18+AMR is competitive to the performances of the related works, which bake rotation invariance directly into their neural network designs.

**AMR usefulness given the prevalence of rotated data**    In an applied scenario, it is not always realistic that all inputs are presented at a random angle. We therefore investigate the usefulness of AMR when the test data consists of a combination of upright (up) and rotated (rot) images. To this end, we compute top-1 test errors on ImageNet of the ResNet family models on rotated and upright inputs separately. We repeat this process for upright training, rotated training and AMR-33 (see Table 4). We then linearly combine up and rot performances to obtain the final performances for mixed datasets consisting of both upright and rotated data. We increase the percentage of rotated data in the test mix until alternative methods (rotated training, AMR-33) start outperforming the default of upright training. We call percentages of parity between methods breakpoints (BP). Unsurprisingly, the BPs for rotated training (30% on average) are much higher

Table 3: Top-1 accuracies on rotated MNIST for ResNet-18 based methods as well as related works, accompanied by ResNet-18 upright top-1 accuracy as a baseline.

| Method | Top-1 Acc. |
| --- | --- |
| ResNet-18 (upright - ceil performance) | 0.996 |
| ResNet-18 | 0.48 |
| ResNet-18 + rotated training | 0.978 |
| ResNet18 + AMR | 0.981 |
| Harmonic Networks Worrall et al. (2017) | 0.983 |
| Ti-pooling Laptev et al. (2016) | 0.988 |
| G-CNNs (P4CNN) Cohen & Welling (2016) | 0.972 |
| RotEqNet (base) Marcos et al. (2017) | 0.989 |

than the ones of AMR-33 (7.5%). The key finding here is that BPs for AMR-33 are all below 10% which shows that only a small portion of the test set needs to be non-upright for AMR to be a worthwhile choice.

Table 4: Top-1 accuracies of ResNets on upright (up) and rotated (rot) ImageNet, accompanied with breakpoints (BP) that signify the share of rotated data in the test set necessary for alternative methods (rotated training, AMR) to outperform upright training.

| | Upright Tr. | | Rotated Tr. | | | AMR-33 | | |
| --- | --- | --- | --- | --- | --- | --- | --- | --- |
| | up | rot | up | rot | BP | up | rot | BP |
| RN-18 | 69.5 | 43.3 | 59.3 | 58.9 | 35.5% | 67.1 | 67.6 | 9.0% |
| RN-50 | 76.5 | 53.7 | 69.2 | 67.3 | 35.0% | 75.1 | 75.5 | 6.1% |
| RN-152 | 77.9 | 55.2 | 73.6 | 73.0 | 19.5% | 76.2 | 76.7 | 7.4% |
| Avg. | 74.6 | 50.7 | 67.4 | 66.4 | 30% | 72.8 | 73.3 | 7.5% |

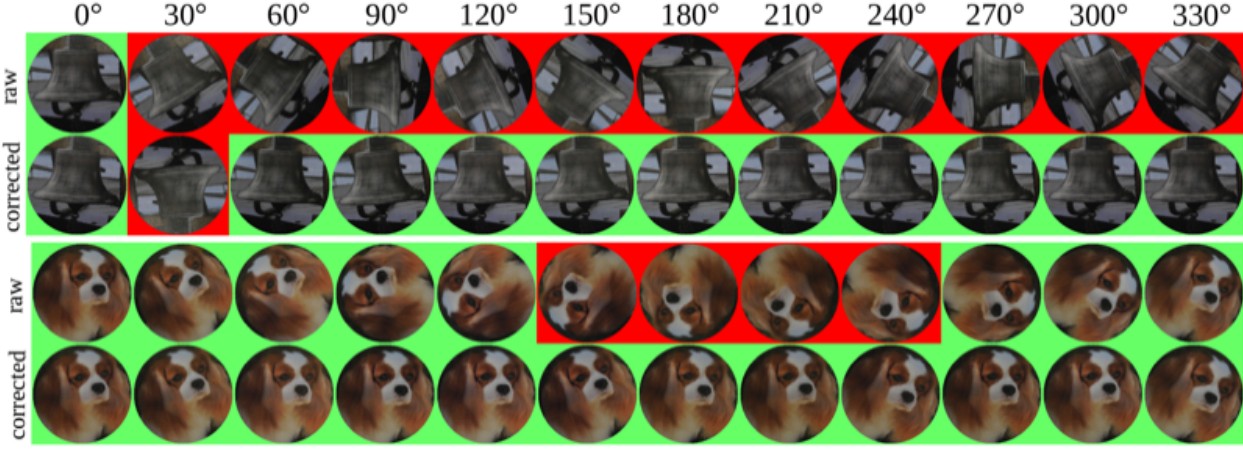

Figure 5: Photographs of two printed ImageNet validation samples taken at 12 different angles. Both samples are shown in their original state (raw) and after the mental rotation step (corrected). The color of the masked-out region indicates if the corresponding image has been correctly classified. The mental rotation and classification steps have been performed by ResNet50 + AMR33.

## 5 Validity of self-supervised training built on artificial rotation

Self-supervised learning based on artificial data modifications always warrants great caution. It is often unclear if the model learns to solve the desired task or if it simply learns to find unintended shortcuts in the self-supervision procedure. In our case, we use a digital rotation algorithm on our input images. While none

| Input | Ground truth | Segmentation | AMR-Segmentation |
|---|---|---|---|

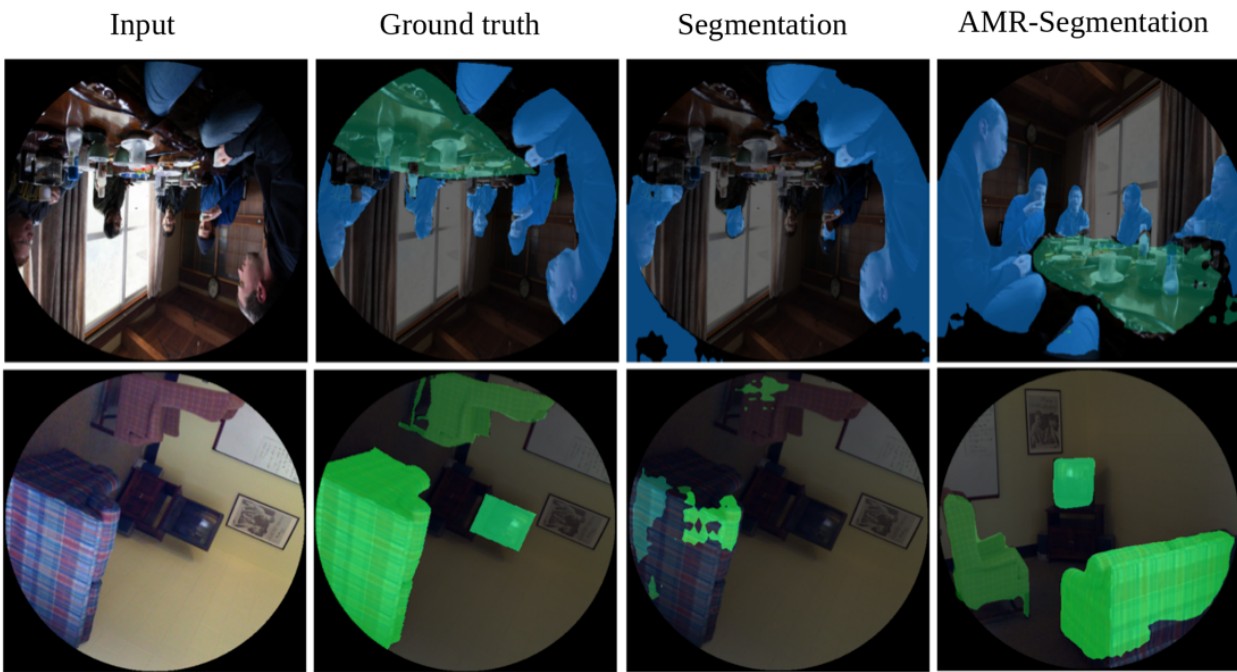

Figure 6: Two rotated examples (rows) from the CoCo validation set with their corresponding ground truth, segmentation output, and AMR-corrected segmentation output (columns). The segmentation on the rotated image exhibits bad performance in both samples. The column AMR-segmentation shows the output for the images that have been (correctly) un-rotated using AMR.

are visible to the human eye, algorithm-specific artefacts are introduced to the rotated images. This raises the question if the AMR module learns to classify the correct rotation angle based on unwanted traces of the rotation algorithm rather than by understanding the contents of the image. To ensure this is not the case, we perform the following ablation study: We print out seven images sourced from different classes from the ImageNet validation set. We then take photos of each of those printouts at twelve different in-plane rotations by physically rotating the print in 30-degree intervals. This way we guarantee the absence of any rotation algorithm artifacts that the model could have learned to use. The images were printed using a Konica Minolta bizhub 450i on maximum resolution with guidelines to enable accurate angular distances (see Figure 8 in the Appendix). The photos were then taken by hand using a Nikon Coolpix P7000 digital camera. Figure 5 shows all twelve re-digitized photos for two cases (raw). The color-coded background indicates if that photo was correctly classified by a standard trained ResNet50 base model (green denotes correct, red an error). We observe a similar effect as with Stanford Cars: Like a car, bells have a very clearly defined upright position. The bell, therefore, is only classified correctly when it is upright. Dogs on the other hand are very variable in appearance (e.g. head turned, laying down etc), thus the dog is only misclassified when it is completely upside down at rotations between 150 and 240 degrees. The second rows (corrected) show the outcome of applying Resnet50 + AMR33 to the above photos. The AMR module is able to correct the orientation of all but one photo. We conclude that it learned to classify the angles by understanding the image contents. While we cannot exclude any learning of artefacts introduced by the self-supervision process, if such are present, they do not hinder the training process from learning transferrable features. We further observe that the rotation correction is much more precise in the bell case than for the dog. This ties in with our assumption that the network's filters are much more precisely tuned to a sharp upright position for the bell compared to the dog. Across all 84 photos, the standard ResNet50 achieves a top-1 classification accuracy of 0.57. ResNet50 + AMR33, on the other hand, achieves a top-1 accuracy of 0.96, showing that the AMR module works properly on all printed images.

# 6 Application to a Novel Downstream Task: Semantic Segmentation

Since they share the same neural network building blocks, the assumption that models for other vision tasks like object detection or semantic segmentation also struggle with rotated inputs suggests itself. In this section, we test this hypothesis and demonstrate how a trained AMR module can be used to easily improve the rotational stability of models for tasks other than classification. We choose semantic segmentation as an example. As the base model, we use a fully convolutional ResNet50 and source the matching pre-trained weights named 'FCN_ResNet50_Weights.COCO_WITH_VOC_LABELS_V1' from torchvision. They have been trained on MSCoCo-Stuff (Lin et al., 2014), with a reduced class set only containing classes that are also available in PascalVOC (Everingham et al.). We again mask the corners of all images. On this masked but upright data, the pre-trained model achieves a mean intersection over union (IoU) of 57.6. Rotating the images causes the mean IoU to drop to 32.7. This confirms our initial conjecture. We now take our ResNet50 + AMR33 which has been trained on ImageNet and use it to perform AMR steps (1) and (2) on CoCo without any additional retraining or modification. The angle-corrected inputs are then fed back into the base semantic segmentation model. This approach yields an IoU of 55.2, showing that AMR also works for semantic segmentation and that a trained AMR module can be easily transferred between similar datasets. Figure 6 shows two examples from the CoCo validation set with their corresponding ground truth, segmentation output, and AMR-corrected segmentation output, visually confirming our findings.

# 7 AMR as a canonicalization function

AMR can be seen as an implementation of a canonicalization function as described by Kaba et al. (2023). A core difference to existing implementations (Mondal et al., 2023) is that our canonicalization function is not learned but hard coded (in the form of the rotation algorithm present in open-CV). It is then individually parameterized per input in a one shot fashion by a learned function - the AMR module. Our architecture encodes a strong inductive bias for in-plane rotations only. This causes the AMR module to be easily trainable within a few epochs, representing a large advantage over training a model to predict canonicalization functions directly, which is empirically difficult to optimize in practice as described by Mondal et al. (2023). However, employing such a strict inductive bias is a trade-off since it will most likely lead to sub-optimal learning in the presence of transformations that can not be encoded within that bias (e.g. rotations off-plane from the image itself). Most datasets such as ImageNet likely contain a wide variety of such perturbations. They can make the self-supervised learning signal more noisy but our results show that they do not critically impair the AMR training

# 8 Limitations and Future Work

A key drawback of AMR is that two forward passes are necessary for inference. This is part of the core design and cannot be changed. It is mitigated partially by the fact that a smaller model can be chosen in conjunction with AMR and still outperform a large model trained with rotational data augmentation due to the inefficiency of that approach resulting in a less costly package even at test time. For example, a forward pass through the AMR module and 2xResNet-50 is 8.905 GFlops, whereas a single forward pass through a ResNet-152 is 11.5 GFlops; still, the AMR-combination outperforms the larger ResNet in this example, see Table 1. With applicability in mind, we opted to focus on 2D in-plane rotations of whole images featuring one dominant object. Our work is, therefore, not suited for cases where multiple objects are individually rotated. This scenario could be addressed by combining a region-proposal-based method such as Faster-RCNN (Girshick, 2015) with AMR at the proposal level. In the real world, 3D objects are rotated in 3D space, which can lead to much more drastic changes in appearance. Extending AMR to this realistic setting (i.e. at the hands of a game engine or interactive learning using robots in the real world) would be a very promising, most natural extension of this work that could lead to vision systems that learn more complete and abstract representations of objects. An exciting future application for AMR models would be reducing the rotational variability of an existing dataset (e.g. ImageNet, by making all appearing objects upright). This would further disentangle the training of upright appearances from rotations which would likely lead to improved training efficiency of base models.

## 9 Conclusions

We have presented AMR, a neuropsychology-inspired approach for handling rotated data in vision systems. We have shown that AMR consistently outperforms rotational data augmentation across different deep architectures and datasets. We have shown the viability of AMR in realistic cases where the data is a mixture of upright and rotated inputs. We further presented a sanity check which confirms that our self-supervised learning setup learns to identify rotations by the content of the images. Lastly, we have shown how a trained AMR module can easily be transferred to another model built for a different task to improve its rotational stability, underpinning its flexibility to be used with any architecture.

## Impact Statement

This paper presents work whose goal is to advance the field of Machine Learning, specifically computer vision. There are many potential societal consequences of our work, none of which we feel must be specifically highlighted here.

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

## A   Are the base network features useful for rotation estimation?

We opt to use the features generated by a base classification network as the input to our AMR module instead of using a distinct specialized network for angle prediction. This allows us to design an AMR module with a very low number of weights such that it is quickly trained. The intuition behind this choice is that the features which are trained for classification also carry valuable information for angle prediction. This should be especially true for the early layers which tend to consist more of lower-level, class-agnostic, features. In this section, we present an ablation study exploring the validity of this assumption. We train AMR modules attached to a ResNet-18 with restricted information flow from the base model to the AMR. First, we train a module and only open the connection from the Stem to the AMR module (see Figure 2). We repeat this for each of the four other connections (Stage 1 - Stage 4). Figure 7 shows the evolution of the top-1 angle classification errors over the training time. The best single source of information is the output of Stage 1. A close second is the module that is only attached to the stem (making it essentially a separate network). As expected are the outputs of the later stages far less informative. We train an additional module with input from Stage 1 plus Stem, it outperforms the other modules which are attached to only one of those quite significantly. This shows that the channels carry some complementary information. Finally, the module attached to the base network at all five connections clearly performs the best. Therefore we can conclude that the features of the base network are helpful for angle prediction and feeding the output of all base network stages into the AMR module is the best design choice.

Table 5: Stanford Cars (top) and Oxford Pet (bottom) top-1 accuracies for all architectures analogous to the one shown above for ImageNet (see Table 1).

Stanford Cars

| Testing | Upright Training | | | Rotated Training | | AMR 300 | | AMR 50 | |
|---|---|---|---|---|---|---|---|---|---|
| | up | rot | % ceil | rot | % ceil | rot | % ceil | rot | % ceil |
| ResNet-18 | 0.854 | 0.163 | 0.19 | 0.412 | 0.48 | **0.812** | **0.95** | 0.800 | 0.94 |
| ResNet-50 | 0.892 | 0.182 | 0.20 | 0.535 | 0.60 | **0.807** | **0.90** | 0.705 | 0.79 |
| ResNet-152 | 0.886 | 0.169 | 0.19 | 0.671 | 0.76 | **0.737** | **0.83** | 0.615 | 0.69 |
| EfficientNet-b0 | 0.825 | 0.140 | 0.17 | 0.687 | 0.83 | **0.820** | **0.99** | 0.785 | 0.95 |
| EfficientNet-b2 | 0.831 | 0.138 | 0.17 | 0.770 | 0.93 | **0.823** | **0.99** | 0.808 | 0.97 |
| EfficientNet-b4 | 0.885 | 0.163 | 0.18 | 0.786 | 0.89 | **0.877** | **0.99** | 0.860 | 0.97 |
| ResNext-50-32x4d | 0.877 | 0.185 | 0.21 | 0.507 | 0.58 | **0.763** | **0.87** | 0.728 | 0.83 |
| ResNext-101-32x8d | 0.885 | 0.179 | 0.20 | 0.577 | 0.65 | **0.725** | **0.82** | 0.666 | 0.75 |
| Average | 0.867 | 0.165 | 0.19 | 0.618 | 0.71 | **0.796** | **0.92** | 0.746 | 0.86 |

Oxford Pet

| Testing | Upright Training | | | Rotated Training | | AMR 1000 | | AMR 150 | |
|---|---|---|---|---|---|---|---|---|---|
| | up | rot | % ceil | rot | % ceil | rot | % ceil | rot | % ceil |
| ResNet-18 | 0.700 | 0.442 | 0.63 | 0.579 | 0.83 | **0.665** | **0.95** | 0.624 | 0.89 |
| ResNet-50 | 0.750 | 0.477 | 0.64 | 0.590 | 0.79 | **0.712** | **0.95** | 0.660 | 0.88 |
| ResNet-152 | 0.752 | 0.460 | 0.61 | 0.579 | 0.77 | **0.697** | **0.93** | 0.633 | 0.84 |
| EfficientNet-b0 | 0.740 | 0.496 | 0.67 | 0.630 | 0.85 | **0.727** | **0.98** | 0.699 | 0.94 |
| EfficientNet-b2 | 0.763 | 0.518 | 0.68 | 0.617 | 0.81 | **0.753** | **0.99** | 0.720 | 0.94 |
| EfficientNet-b4 | 0.735 | 0.515 | 0.70 | 0.605 | 0.82 | **0.719** | **0.98** | 0.686 | 0.93 |
| ResNext-50-32x4d | 0.743 | 0.489 | 0.66 | 0.579 | 0.78 | **0.711** | **0.96** | 0.679 | 0.91 |
| ResNext-101-32x8d | 0.748 | 0.469 | 0.63 | 0.645 | 0.86 | **0.712** | **0.95** | 0.656 | 0.88 |
| Average | 0.741 | 0.483 | 0.65 | 0.603 | 0.81 | **0.712** | **0.96** | 0.670 | 0.90 |

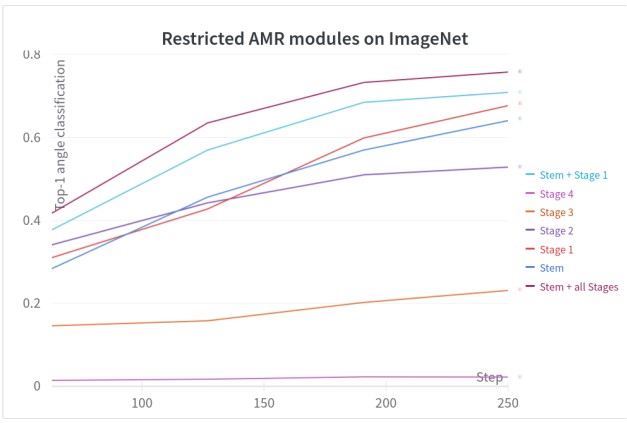

Figure 7: Top-1 angle accuracies for ResNet-18+AMR with the information flow from the base network restricted. Legend indicates which channels (see Figure 2 in the main manuscript) from the base network are open.

Table 6: Top-1 and top-5 classification accuracies of the angle-classifiers, contrasted with the corresponding gains in top-1 image-classification accuracies achieved

| | Angle Classificaton | | | | Base Classification | | AMR Classification | | | |
| --- | --- | --- | --- | --- | --- | --- | --- | --- | --- | --- |
| | long training (LT) | | short training (ST) | | up | rot | LT | % ceil | ST | %ceil |
| | top-1 | top-5 | top-1 | top-5 | | | | | | |
| ImageNet | 0.834 | 0.999 | 0.72 | 0.889 | 0.730 | 0.501 | 0.716 | 98 | 0.708 | 97 |
| Stanford Cars | 0.56 | 0.932 | 0.146 | 0.547 | 0.867 | 0.165 | 0.769 | 92 | 0.746 | 86 |
| Oxford Pet | 0.756 | 0.985 | 0.07 | 0.458 | 0.712 | 0.483 | 0.712 | 96 | 0.670 | 90 |

## B  Accuracies of the angle classifiers

The goal of AMR is to boost the classification accuracy of rotated images. Therefore is the classification accuracy of the angle-classifiers only of secondary interest as a means to an end. However, it is still interesting to investigate how well the angle classifiers work and how their performance correlates with the overall image classification performance. Table 6 shows the accuracies of the different AMR-modules averaged across all our CNN architectures. We observe a significant difference in angle classification accuracies for the longer and shorter training regimes that only partly translates to the downstream image classification performances. Furthermore seem the top-5 angle classification accuracies much more indicative of the downstream performance compared to the top-1 ones. This indicates that the base networks exhibit some robustness to small rotations and the input only has to be upright "enough". This finding is further supported by the observation that the short training AMR classifier for Oxford Pet, which exhibits very poor accuracies still yields a significant increase in downstream image classification accuracy, from 48.2 to 67 percent.

## C  Reproducibility notice

This section provides all the necessary information to reproduce every result presented in this work. Our experiments were run on machines with 4xTesla V100-SXM2-32GB or 4xNVIDIA Tesla T4 16GB. When running on a single GPU we recommend the linear learning rate scaling rule Goyal et al. (2017).

**Data**
ImageNet, Stanford Cars and Oxford Pet are public datasets and can be sourced from their original authors. Our photographed-at-an-angle versions of ImageNet images can be downloaded here: `removedforreview`

**Code**
Our code is publicly available on GitHub at `removedforreview`

**Hyperparameters**
We logged hyperparameters as well as relevant metrics of each run to `wandb.ai`. The easiest and safest way to exactly replicate a run is to first check out the git state and then use the run command given in the overview tab of each run. The logs of each training and evaluation run can be found under the URLs shown in Table 7.

**Model weights**
The model weights for each trained model we use here can be obtained using the following link `removedforreview`.

## D  Re-digitized Images

Figure 8 shows one example of the printouts that were used to create the manually rotated images for Section 5. The additional helper lines were printed to facilitate the creation of photographs of the printout at exactly the twelve desired angles. Figure 9 shows the re-digitized images at an angle (top rows) and after AMR (bottom rows) with their classifications color coded analogous to Figure 5 but for all seven test images.

Table 7: Links to `wandb.ai` projects containing hyperparameters and relevant metrics of all training and evaluation runs created for this work.

| **ImageNet CNN** | |
| --- | --- |
| *Training* | |
| Base models | `https://wandb.ai/removedforreview` |
| AMR modules | `https://wandb.ai/removedforreview` |
| *Evaluation* | |
| Base models | `https://wandb.ai/removedforreview` |
| BM + AMR | `https://wandb.ai/removedforreview,` |
| | `https://wandb.ai/removedforreview` |
| **ImageNet ViT** | |
| *Training* | |
| Base models | `https://wandb.ai/removedforreview` |
| AMR modules | `https://wandb.ai/removedforreview` |
| *Evaluation* | |
| Base models | `https://wandb.ai/removedforreview` |
| BM + AMR | `https://wandb.ai/removedforreview` |
| **Stanford Cars** | |
| *Training* | |
| Base models | `https://wandb.ai/removedforreview` |
| AMR modules | `https://wandb.ai/removedforreview` |
| *Evaluation* | |
| Base models | `https://wandb.ai/removedforreview` |
| BM + AMR | `https://wandb.ai/removedforreview` |
| **Oxford Pet** | |
| *Training* | |
| Base models | `https://wandb.ai/removedforreview` |
| AMR modules | `https://wandb.ai/removedforreview` |
| *Evaluation* | |
| Base models | `https://wandb.ai/removedforreview` |
| BM + AMR | `https://wandb.ai/removedforreview` |
| **MNIST** | |
| *Training* | |
| Base models | `https://wandb.ai/removedforreview` |
| AMR modules | `https://wandb.ai/removedforreview` |

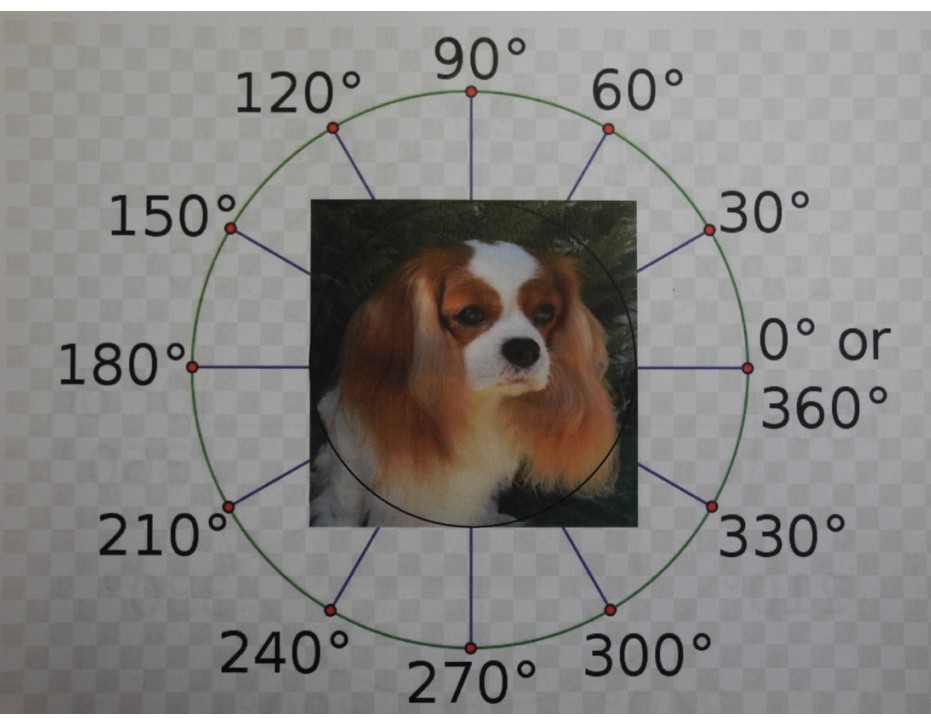

Figure 8: Example printout featuring helper lines to facilitate photographs at exact angles.

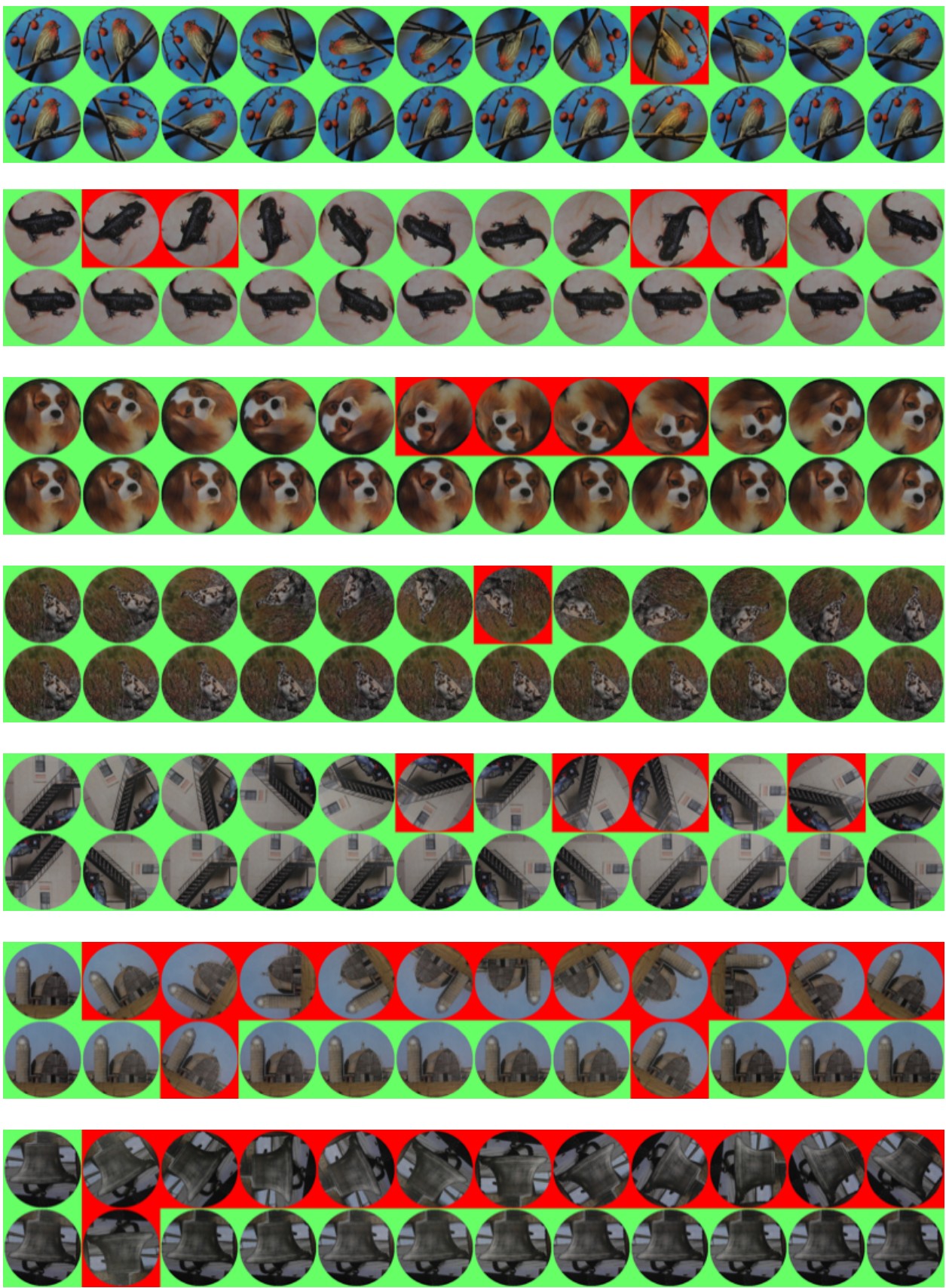

Figure 9: All re-digitized photos raw (top rows) and after AMR (bottom rows) with their corresponding classifications color coded.

