# OpenReview forum: "Efficient Rotation Invariance in Deep Neural Networks through Artificial Mental Rotation"
_TMLR — Rejected by TMLR_

### Review · Reviewer_iCG2 · 2024-12-08

**Summary Of Contributions:**

Motivated by the difficulty of generalizing vision models to images with out-of-distribution planar rotations, the authors introduce a pre-processing canonicalization step. They train a model to classify the angle at which an image has been randomly augmented. During testing, the rotation prediction is used to re-canonicalize the image before it is passed to a task-specific model. The authors demonstrate that a task-specific model combined with their preprocessing outperforms the task-specific model trained with 360-degree random rotations on several rotated-image evaluations.

**Audience:**

Yes

**Claims And Evidence:**

No

**Requested Changes:**

1. The authors must appropriately qualify their claims as detailed above.
2. The related-work section should be expanded to: A) make clear that the authors do not "introduce the concept of mental rotation to deep learning"; B) include discussion of works that learn a canonicalization function, as mentioned in W.5; and C) clarify how the proposed method differs from prior work.
3. The authors should compare against baseline methods in the "large datasets" evaluations, including at least Boominathan et al. (2016), Marcos et al. (2017) (which was the best-performing method in the Rotated MNIST evaluation), and some variant of Spatial Transformer Networks [4], each with comparable architecture sizes.

Additional:
1. The PDF is excessively large due to the authors' inclusion of a handful of large, uncompressed images. To improve accessibility, the authors should appropriately resize and compress their images.
2. To improve clarity, `\citep` should be used instead of `\citet` throughout when the citation is not the object of the sentence. See the TMLR template for examples.
3. The caption and axis label of Figure 1 seems as though they should read "Top-1 Accuracy." The authors should verify this, and the fill-between shading in Figure 1 should also be explained.
4. The claim that the use of rotational augmentation results in "slower training" does not appear to be backed by Figure 1, which seems to show two parallel curves. What evaluation data is used in Figure 1? This and the claim should be clarified.
5. The authors write that "transformers Vaswani et al. (2017) (a.k.a. neural fast weight programmers Schmidhuber (1992); Schlag et al. (2021)) inherently integrate translational invariance into their design," but it should be clarified that, unlike CNNs, transformers (and ViTs, see [7] for a detailed discussion) are not translationally invariant.
6. In what setting does rotational augmentation result in "lower final performance"? This runs counter to established tradition that adding small rotational augmentations increases model performance. Are the authors applying the augmentation in Figure 1 "in accordance with the training recipes for the torchvision base models"? The claim should be clarified.
7. The key motivational claims "For rotations, this is not the case and both methods perform very poorly when facing inputs at an unusual angle" and that it "causes serious issues in applications where rotated inputs are common" should be backed by citations.
8. The authors write "Marcos et al. (2017) on the other hand proposed to rotate the convolutional filters instead of the representations," but the statement is also true of Cohen & Welling (2016), which the sentence is positioned in contrast to. The authors should review their discussion of related work.
9. In the following, the authors must clarify they are referring to rotated versions of the benchmarks, and not the benchmarks themselves: "(c) we extensively test the merits of AMR on ImageNet, Stanford Cars, and Oxford Pet, and conclude that it significantly outperforms data augmentation, the current state-of-the-art, (d) we present AMR results on MNIST to enable the comparison with computationally expensive alternatives".
10. The claim "(e) we study the viability of AMR in a scenario where only parts of the test data are rotated" should read *portions* instead of *parts* to improve clarity. As it is now written, it lends the impression that parts of individual samples are altered.
11. The authors claim "(f) we present comprehensive ablation studies proofing that our trained AMR modules work in practice on synthesized as well as naturally rotated data," but they perform no evaluation on naturally rotated data. The claim should be removed.
12. The author's assertion regarding prior work that "All these methods share the key drawback that their memory footprint grows linearly with the angular resolution" is not correct. Angular resolution is not applicable to all methods listed, such as Worrall et al. (2017). The authors also note of Marcos et al. (2017) that they "rotate the filters of a CNN and then apply spatial and orientation pooling to reduce and merge the resulting features," which introduces only a constant memory overhead. The authors must qualify their claim.
13. The claim "However, data augmentation is not efficient because different appearances of the same object are learned independently." should be supported with a citation.
14. The claim "There is no need to modify the BM in any way," is confusing given that modifications are necessary to integrate the AMR module at five different feature depths. This should be clarified.
15. "% ceil" in Tables 1 and 2 should be multiplied by 100.
16. The claim "It is mitigated partially by the fact that a smaller model can be chosen in conjunction with AMR and still outperform a large model trained with rotational data augmentation due to the inefficiency of that approach resulting in a less costly package even at test time." does not seem to have been evaluated, and should be removed otherwise.
17. The authors should specify baseline architectures in the MNIST evaluation.
18. The statement "In the real world, 3D objects are rotated on two axes," should read three axes.
19. The phenomena observed that the "ResNe(x)t models showed a decline in performance when approaching 0 degrees, while EfficientNets do not suffer from this effect" is interesting. However, the authors' explanation that the models "can learn very task-specific filters in early layers that only retain information used for classification" does not seem related. The section in the appendix where the authors "investigate this phenomenon further" should be removed if the authors are not able to offer a potential explanation.
20. The authors should include angle-classification performance metrics for all experiments.

[7] Rojas-Gomez, R. A., Lim, T. Y., Do, M. N., & Yeh, R. A. (2024). Making vision transformers truly shift-equivariant. In Proceedings of the IEEE/CVF Conference on Computer Vision and Pattern Recognition (pp. 5568-5577).

**Strengths And Weaknesses:**

Strengths:
1. In contrast to approaches that introduce rotational-invariance inductive biases into the target network, the proposed method can be applied ad-hoc and may have greater applicability in some domains.
2. The paper's ideas and methodology are generally clear.
3. The authors' attempt to ensure that the proposed network is not learning to take shortcuts introduced by the rotation algorithm is commendable.

Weaknesses:
1. The authors do not "introduce the concept of mental rotation to deep learning," which has been featured in a number of prior works, including those cited: A) The authors write of Fang et al. (2020), "In the space of 3D vision a mental rotation-based approach achieved state-of-the-art performance for rotated point cloud classification." B) Boominathan et al. (2016) is also self-described as being "primarily inspired from the neuro-scientific concept of mental rotations."
    Other "deep learning" works in which the proposed method was explicitly likened to mental rotation include [1], [2], and [3].
2. The claim "This shows that AMR not only exhibits the best performance on large datasets but also achieves state-of-the-art performance on a small dataset amongst highly specialized methods." is not supported by the evaluations. No comparisons were performed on "large datasets," and in the only evaluation made, on Rotated MNIST, the proposed approach is outperformed by three of the four baseline methods.
3. The authors' printout experiment does not address the question of whether the model found "unintended shortcuts in the self-supervision procedure." It can be true that the model learned both transferable features and domain-specific shortcuts. The ensuing discussion must be appropriately qualified.
4. The claim that the baseline methods are "much narrower in scope" has not been established. In fact, the proposed method may be more limited as it assumes a canonical orientation exists, and it would likely not be useful on astronomical (Dieleman et al. (2015)) or aerial (Marcos et al. (2017)) imagery.
5. Relatedly, the authors make no mention of works that automatically learn a canonicalization function, such as Spatial Transformer Networks [4] and related works [3], [5], and [6]. Rather than canonicalizing only rotation, [4] learns to produce a more-general instance-specific affine transformation.

[1] Efe, U., Ince, K. G., & Alatan, A. (2021). DFM: A Performance Baseline for Deep Feature Matching. In Proceedings of the IEEE/CVF Conference on Computer Vision and Pattern Recognition (pp. 4284-4293).

[2] Singhal, U., Esteves, C., Makadia, A., & Yu, S. X. (2023). Learning to Transform for Generalizable Instance-wise Invariance. In Proceedings of the IEEE/CVF International Conference on Computer Vision (pp. 6211-6221).

[3] Kaba, S. O., Mondal, A. K., Zhang, Y., Bengio, Y., & Ravanbakhsh, S. (2023, July). Equivariance With Learned Canonicalization Functions. In International Conference on Machine Learning (pp. 15546-15566). PMLR.

[4] Jaderberg, M., Simonyan, K., & Zisserman, A. (2015). Spatial Transformer Networks. Advances in Neural Information Processing Systems, 28.

[5] Esteves, C., Allen-Blanchette, C., Zhou, X., & Daniilidis, K. (2018, February). Polar Transformer Networks. In International Conference on Learning Representations.

[6] Tai, K. S., Bailis, P., & Valiant, G. (2019, May). Equivariant Transformer Networks. In International Conference on Machine Learning (pp. 6086-6095). PMLR.

---

> ### Author Response · Authors · 2024-12-25
> **Intended changes to manuscript**
>
> Dear reviewer, Thank you for the extensive and insightful review. This post details how we aim to incorporate your feedback into our paper. Please feel free to respond directly should you disagree with some of our planned actions.
>
> Regarding the main weaknesses:
> 1. Regarding the "introduction" of mental rotation to deep learning, we re-phrase our claim to our work being mental rotation inspired rather than introducing mental rotation to DL.
> 2. We do a very bad job communicating why we call the other methods which are benchmarked on MNIST "very specialized". They are not specialized towards MNIST itself but are highly specialized in the sense that they spend an enormous amount of computation to achieve rotational invariance and are therefore only properly applicable to smaller datasets. We rephrase our argument in the paper to more clearly state this.
> 3. It is true that our experiment only confirms that transferrable features are learned and not the absence of domain-specific shortcuts. We change our wording accordingly.
> 4. We will amend our related works with the suggested works and an accompanying discussion.
>
>
> Regarding the additional requested Baselines. We will add the following:
> - A comparison with Boominathan et al. (2016), by training their angle classifier on ImageNet and combining it with our trained ResNet-50.
> - A comparison with Marcos et al. (2017) as implemented here: https://github.com/COGMAR/RotEqNet. This method has a symmetry of 17 angles - which leads to a 17-fold memory footprint. Hence it will scale very poorly to ImageNet-sized datasets. This is the whole point we (admittedly poorly) tried to make by calling these methods "specialized". Nevertheless, we will try to train a RotEqNet on ImageNet.
>
>
> Learning a canonicalization function as [4][5][6] is not in the scope of our work. Moreover, none of the mentioned works reports results on any problem which is even close to ImageNet-sized. So we feel it is "a bridge too far" to ask us to undertake this scaling, which is likely very involved, and maybe even impossible on current hardware as part of a baseline comparison.
>
> The suggestions in the additional remarks we will incorporate in the paper as suggested unless we mention them here:
>
> - Nr. 11 By naturally rotated data we mean non-algorithmically, as we did for our ablation study. We re-phrase the claim accordingly
> - Nr. 12 Thanks for pointing out the wrong claim about Worrall et al. (2017), revise the section accordingly. Marcos et al. (2017) apply the filter at N rotations before they apply pooling. This causes a memory footprint proportional to N, hence this point still stands.
> - Nr. 14 Copying features out of an architecture does not constitute a change to the architecture itself in our view. We add a sentence that we copy data out of the BM
> - Nr. 16 1xAMR + 2x ResNet-50 is less flops than 1x ResNet-152 yet it has better performance - we will add this as an explicit example.
> - Nr. 18 in three dimensions translation has 3 axes but rotation only has 2 independent ones
>
>
> P.S. Please excuse the late response I was fully absorbed by another project until the holidays

---

> > ### Author Response · Authors · 2024-12-27
> > **Boominathan et al. (2016) baseline.**
> >
> > We have re-implemented a binary Upright/Not-upright classifier according to the specifications of Boominathan et al. (2016). Not everything was properly documented so we had to make some assumptions regarding pooling sizes/strides to make their fcn-less design work.
> >
> > Unfortunately, the method of Boominathan et al. (2016) is not able to learn anything beyond random accuracy on ImageNet.
> >
> > Feel free to inspect the implementation here:
> > https://pastebin.com/FyYfBGLV
> >
> > We will move on and attempt to train an RotEqNet on ImageNet in the coming days.

---

> > > ### Comment · Reviewer_iCG2 · 2024-12-30
> > >
> > > As the authors have responded to only a subset of comments relating to correctness issues, it is unclear what changes they intend to make. Accordingly, outstanding concerns will be reraised following the submission of revisions.
> > > 1. The authors' statement that methods "[l]earning a canonicalization function as [4][5][6] [are] not in the scope of [their] work" is not correct. The referenced works are a generalization of the proposed method, subsuming its functionality. All claims relating to "state of the art" must be removed if the authors are unwilling to correctly contextualize their work. The authors' statement that "none of the mentioned [canonicalization] works reports results on any problem which is even close to ImageNet-sized" is confusing as the authors of [4] also base an experiment off of a model pre-trained on ImageNet. That previous work has not also evaluated on a rotated version of ImageNet only highlights the authors' choice of a nonstandard evaluation setting.
> > > 2. Nr. 12: The authors' claim regarding the memory complexity of Marcos et al. (2017) relates to implementation details and not material method limitations. Do the authors contest that a convolution followed by a pooling requires only a constant factor over the size of the convolution input and pooling output? The reduction of memory-usage requirements is a contribution of Marcos et al. (2017) and should be properly acknowledged.
> > > 3. Nr. 18: The authors' statement that "in three dimensions . . . rotation only has *2* independent [dimensions]" (emphasis added) is not correct. The related statement in Section 6 should either be removed or backed by a citation.
> > > 4. The implementation the authors have provided in [7] neither appears to be a complete reproduction of Boominathan et al. (2016) nor has a comparable architecture size. Where is the Gaussian-process-based regressor? Performance can not be measured by absolute classification accuracy, as relative scores are used for rotation selection. It's not clear how the authors have determined that "the method of Boominathan et al. (2016) is not able to learn anything beyond random accuracy on ImageNet," but, if the authors are unable to reproduce the results, they are asked to contact the paper's authors for additional details. Boominathan et al. (2016) found that fitting even an SVM classifier on HoG features outperformed a random baseline. The motivational and technical differences between the methods are limited, so it is of greater importance that the authors perform a fair comparison.
> > >
> > > [7] https://pastebin.com/FyYfBGLV
> > >
> > > [7A] https://archive.ph/i8iha

---

> ### Author Response · Authors · 2025-01-12
> **Revised Manuscript**
>
> We have re-shaped the paper to include references and a discussion of missing related works with a focus on canonicalization functions, including revised novelty claims. We have sharpened the focus of our contribution towards the practical viability of our approach we achieve through the disentanglement of our method with the base architecture instead of SoTA comparisons on small datasets (i.e. MNIST).

---

> > ### Comment · Reviewer_iCG2 · 2025-01-12
> >
> > The authors' revisions are appreciated.
> >
> > Would the authors please clarify whether they intend to submit further revisions? The posted changes do not appear to include the comparisons promised in the rebuttal.

---

> > > ### Author Response · Authors · 2025-01-12
> > >
> > > We have re-focused our message on the main point that a method which aims for practical relevance needs to be disentangled and seamlessly integrate with the ever-evolving Zoo of growing neural network architectures. We no longer claim any SoTA status only that we are competitive on a common dataset such as MNIST while fulfilling the abovementioned criterion.
> > > Hence we moved away from adding any additional comparisons

---

> > > > ### Comment · Reviewer_iCG2 · 2025-01-13
> > > >
> > > > It is disappointing that the authors have decided to remove performance claims rather than appropriately contextualize their work by adding important comparisons.
> > > >
> > > > First, reraised, are the following unaddressed concerns:
> > > > 1. W.1: In their rebuttal, the authors stated their intention to "re-phrase [their] claim to [their] work being mental rotation inspired rather than introducing mental rotation to DL," but the revised text still claims the authors "introduce artificial mental rotation."
> > > > 2. W.2: It is not appropriate to claim that "the performance of ResNet18+AMR is competitive to the performances of the related works" when it is outperformed by up to 0.8% by three out of four baselines. To contextualize that difference, the authors claim "the performance of AMR *far superior* to rotated training" (emphasis added) when it differs by only 0.3%. The claim that "AMR remains the most potent way of addressing rotations" must also be removed as this statement is not supported.
> > > > 3. W.3: Regarding the printout experiment, while the authors acknowledged in their rebuttal that the "experiment only confirms that transferrable features are learned and not the absence of domain-specific shortcuts," they have not appropriately qualified their discussion in the revision ("We can therefore conclude that it learned to classify the angles by understanding the image contents rather than relying on artifacts introduced by the self-supervision process."). The authors must make explicit what the experiment does and does not show.
> > > > 4. W.5: The discussion in "5.1 AMR as a canonicalization function" should be expanded to include discussion of limitations of the proposed approach relative to existing canonicalization works. If ImageNet contains planar rotations, the authors' "hard coded" canonicalization method will presumably learn sub-optimal rotational biases. Additionally, the statement that the authors' "design has the advantage that it does not include an optimisation to find the needed canonicalization function for each input," appears misleading and should be removed as the work it is positioned in contrast to, Kaba et al. (2023), experiments with both direct and optimization-based approaches.
> > > > 5. Nr. 3: What interval is used for the "confidence band"?
> > > > 6. Nr. 4: The text of Figure 1 was modified from "revealing a training slow-down" to "revealing lower performance," but the introduction still states that rotational augmentation is "an inefficient method because reducing the sample efficiency and consequently resulting in
> > > > slower training(see Figure 1)." This does not appear to be backed by Figure 1, which seems to show two parallel curves. What evaluation data is used in Figure 1? This is not discussed in the paper and the figure is otherwise not interpretable.
> > > > 7. Nr. 7: The authors have not added citations for the key motivational claim that "For rotations, this is not the case and both methods perform very poorly when facing inputs at an unusual angle." It has not been demonstrated that "CNNs and vision transformers (ViTs) both perform poorly on rotated inputs." In their revision, the authors add that, for "foundation models," "a decoupled method facilitating the addition of rotation invariance is needed to solve the problem of degraded performance in the presence of in-plane rotation efficiently." However, no foundation models are included in the evaluation, nor foundational citations provided. These claims must otherwise be removed.
> > > > 8. Nr. 9: In the following, the authors have not clarified they are referring to rotated versions of the benchmarks and not the benchmarks themselves:
> > > >    > (c) we extensively test the merits of AMR on ImageNet, Stanford Cars, and Oxford Pet, and conclude that it significantly outperforms data augmentation, the current state-of-the-art, (d) we present AMR results on MNIST to enable the comparison with computationally expensive alternatives.
> > > >
> > > >    Confusingly, in the revision, the authors have instead extended the phrase to read they "test the *real-world* merits on ImageNet . . ." (emphasis added). It must be clarified throughout that the authors do not evaluate on the benchmarks themselves.
> > > > 9. Nr. 14: The authors do not appear to have added the sentence referenced in their rebuttal.
> > > > 10. Nr. 16: The authors claim in their rebuttal that "1xAMR + 2x ResNet-50 is less flops than 1x ResNet-152 yet it has better performance - we will add this as an explicit example," but they do not appear to have included quantitative results in their revision. The text should be otherwise removed.
> > > > 11. Nr. 17: The authors have not specified the baseline architectures in the MNIST evaluation.
> > > > 12. Nr. 19: The authors do not appear to have made any changes to this discussion.
> > > > 13. Nr. 20: The authors still do not appear to have included angle-classification performance metrics in any evaluation.

---

> > > > > ### Comment · Reviewer_iCG2 · 2025-01-13
> > > > >
> > > > > New concerns:
> > > > >
> > > > > 14. Nr. 21: In the revision, the authors added that they "test the real-world merits of AMR," that the focus of AMR is "on large datasets and real-world tasks," and that AMR "generally shows excellent performance in practically relevant tasks." However, it must be clarified that the proposed method is only quantitatively evaluated in synthetic settings. Reference to "real-world" performance should be removed.
> > > > > 15. Nr. 22: The authors have introduced a number of typos in their revision, contrasting with the quality of the original text. Examples: 1) "This demands methodological simplicity with to low development overhead and fast execution w.r.t. runtime on top of excellent performance on various downstream vision tasks (e.g. classification or segmentation)." 2) "While improving rotational stability do data agumentation approaches not tackle the issue on a fundamental level."
> > > > > 16. Nr. 23: This was not noticed previously, but the authors seem to have forgotten to add an introduction `\section` to the paper and are recommended to do so.

---

> ### Author Response · Authors · 2025-01-13
>
> We thank the Reviewer for his continued detailed remarks which we truly believe helped make our paper significantly better. Very much appreciated! We apologize for the overly brief comments accompanying the last submission, which was inadequate. In the list below we address all concerns point by point (we will do another revision tomorrow at noon CET solely focused on typos and text quality, we already submit now to ensure the tool Is still open):
> 1. *W.1: In their rebuttal, the authors stated their intention to "re-phrase [their] claim to [their] work being mental rotation inspired rather than introducing mental rotation to DL," but the revised text still claims the authors "introduce artificial mental rotation."*
>
> Yes, this has been an unfortunate formulation as we dubbed our method AMR (which we want to claim, not the inspiration by the physiological phenomenon).  We disentangled these two layers of meaning now in the following way (starting to quote from the previous paragraph that explains the physiological phenomenon): “[…]This concept of mental rotation has been of inspiration to several computer vision methods \cite{…}.
> In this paper, we introduce the Artificial Mental Rotation (AMR) method that separates the finding of the angle of rotation of a given input from subsequent rotating it back to its canonical appearance before further processing, thus performing an artificial version of mental rotation […]”.
>
> 2.	*W.2: It is not appropriate to claim that "the performance of ResNet18+AMR is
> competitive to the performances of the related works" when it is outperformed by up to 0.8% by three out of four baselines. To contextualize that difference, the authors claim "the performance of AMR far superior to rotated training" (emphasis added) when it differs by only 0.3%. The claim that "AMR remains the most potent way of addressing rotations" must also be removed as this statement is not supported.*
>
> The claim that it is far superior has been plain wrong as you pointed out – we changed it and added our interpretation: “Similar to the larger datasets above is the performance of AMR superior to rotated training, however only by a small margin. This makes sense intuitively since for such a simple dataset the reduced sample efficiency of rotated training plays a small role.”
> By competitive de do not claim that it is the best but it is in the same "ballpark".
>
> Additionally, we clarified that our findings are constrained to the datasets we test on: “Our core findings are replicated on both datasets: Rotating the images reduces all the models' performances and AMR remains the most potent way of addressing rotations on these datasets”.
>
> 3.	*W.3: Regarding the printout experiment, while the authors acknowledged in their rebuttal that the "experiment only confirms that transferrable features are learned and not the absence of domain-specific shortcuts," they have not appropriately qualified their discussion in the revision ("We can therefore conclude that it learned to classify the angles by understanding the image contents rather than relying on artifacts introduced by the self-supervision process."). The authors must make explicit what the experiment does and does not show.*
>
> In our understanding, we did no longer claim the absence of domain-specific shortcuts. However, the word “rely” did a lot of heavy lifting in that formulation. We moved to a more clear and explicit formulation:
> “We conclude that it learned to classify the angles by understanding the image contents. While we cannot exclude any learning of artifacts introduced by the self-supervision process, if such are present, they do not hinder the training process from learning transferrable features.”
>
> Continued below

---

> ### Author Response · Authors · 2025-01-13
>
> 5.	*Nr. 3: What interval is used for the "confidence band"*?
>
> We have replaced this figure altogether (see our comment to concern 6.).
>
> 6.	*Nr. 4: The text of Figure 1 was modified from "revealing a training slow-down" to "revealing lower performance," but the introduction still states that rotational augmentation is "an inefficient method because reducing the sample efficiency and consequently resulting in slower training(see Figure 1)." This does not appear to be backed by Figure 1, which seems to show two parallel curves. What evaluation data is used in Figure 1? This is not discussed in the paper and the figure is otherwise not interpretable.*
>
> We see that this figure was not the best visualization to support our point, hence we replaced it by polar plots showing differences between upright training and rotated training in final accuracy when the training time is kept the same.
>
> 7.	*Nr. 7: The authors have not added citations for the key motivational claim that "For rotations, this is not the case and both methods perform very poorly when facing inputs at an unusual angle." It has not been demonstrated that "CNNs and vision transformers (ViTs) both perform poorly on rotated inputs."
> In their revision, the authors add that, for "foundation models," "a decoupled method facilitating the addition of rotation invariance is needed to solve the problem of degraded performance in the presence of in-plane rotation efficiently." However, no foundation models are included in the evaluation, nor foundational citations provided. These claims must otherwise be removed.*
>
> We have replaced Figure 1 with a – in our opinion – much more suitable one that shows the poor performance of CNNs and ViT on rotated data, and added relevant citations for both architectures. As the claims are in our perception well-known in the field (as they stem from the architectural inductive biases of CNNs and ViT, respectively), there are possibly other references making similar claims, hence we suggest both as examples.
>
> Regarding foundation models, it is correct that we do not experiment with them. We however see it as a prime advantage of our method that, because it does not induce any change in the underlying vision base model (put differently: it disentangles the equivariance property from the vision backbone), it CAN be used (in practice) with a (hypothetically large) foundation model that would be prohibitively costly to retrain to harden it against rotations. Put yet another way: AMR lets you bring equivariance along to any (e.g., new, future) DNN architecture. This gives you free choice of backbone without developmental hassle. Foundation models are brought into the discussion as it is directly clear to readers in the field that those are an extreme example: such models are either rotation equivariant by design, or need a method with AMR’s “decoupled equivariance” property to be made so without investing prohibitive additional training effort.
>
> We added a reference to SAM to make the example complete. Could we make the point clear or do you see a need for further elaboration in the manuscript?
>
>
> 8.	*Nr. 9: In the following, the authors have not clarified they are referring to rotated versions of the benchmarks and not the benchmarks themselves:
> (c) we extensively test the merits of AMR on ImageNet, Stanford Cars, and Oxford Pet, and conclude that it significantly outperforms data augmentation, the current state-of-the-art, (d) we present AMR results on MNIST to enable the comparison with computationally expensive alternatives.
> Confusingly, in the revision, the authors have instead extended the phrase to read they "test the real-world merits on ImageNet . . ." (emphasis added). It must be clarified throughout that the authors do not evaluate on the benchmarks themselves.*
>
> To clarify this we rephrased the mentioned sentence to “…real-world merits of AMR on rotated versions of ImageNet,...”. Additionally, we explicitly stated that we use rotated versions of the datasets again in the Experiments section. (We never meant to refer to the benchmark procedures, but to the datasets – sorry for the ambiguities.)
>
> 9.	*Nr. 14: The authors do not appear to have added the sentence referenced in their rebuttal.*
>
> Good catch, thank you - this slipped our attention between some re-writes, please excuse the mistake. We have added the following statement:
> “[…] BM can generally be sourced in a fully (pre-)trained form. However, we do copy features out of the BM at various stages, in cases where this is not possible e.g. when using a BM hidden behind an API the AMR module has to be designed as a stand-alone network (this would be equivalent to Stem in Chapter ref{chp_stages}). Additionally, it requires […]”
>
> Continued Below

---

> > ### Author Response · Authors · 2025-01-13
> >
> > 10.	*Nr. 16: The authors claim in their rebuttal that "1xAMR + 2x ResNet-50 is less flops than 1x ResNet-152 yet it has better performance - we will add this as an explicit example," but they do not appear to have included quantitative results in their revision. The text should be otherwise removed.*
> >
> > We have added the following explanatory sentence: “[…] For example, a forward pass through the AMR module and 2xResNet-50 is 8.905 GFlops, whereas a single forward pass through a ResNet-152 is 11.5 GFlops; still, the AMR-combination outperforms the larger ResNet in this example, see Table \ref{tbl_imagenet}. […]” to “limitations and future work” to address the issue of the flops explicitly. The respective quantitative results are listed in Table 1, and we added a cross reference to it.
> >
> > 11.	*Nr. 17: The authors have not specified the baseline architectures in the MNIST evaluation.*
> >
> > We write “[…] we present the performances of ResNet-18 \citep{he2016deep}, ResNet-18 + rotated training and ResNet18+AMR on MNIST (see Table ref{tbl_mnist}) […]”, and have now added the reference to He et al. to specify exactly which ResNet we refer to. Is this what you meant?
> >
> > 12.	*Nr. 19: The authors do not appear to have made any changes to this discussion.*
> >
> > We have removed the chapter in question in the Appendix due to its lack of meaningful results. We originally kept it to spur further investigation into this rather strange behavior but agree that the results lack substance at the moment.
> >
> > 13.	*Nr. 20: The authors still do not appear to have included angle-classification performance metrics in any evaluation.*
> >
> > We newly added Section B in the Appendix, which contains reports on the classification accuracies of the angle classifications. We first suspected that the angle-classification might be a means to an end of little interest; however, our short investigation shows an interesting result: that the angle classification accuracy is less clearly correlated to the overall improvement trough AMR than we intuitively expected. So, this suggestion was again highly appreciated.
> >
> > 14.	*Nr. 21: In the revision, the authors added that they "test the real-world merits of AMR," that the focus of AMR is "on large datasets and real-world tasks," and that AMR "generally shows excellent performance in practically relevant tasks." However, it must be clarified that the proposed method is only quantitatively evaluated in synthetic settings. Reference to "real-world" performance should be removed.*
> >
> > Thanks for raising this point, that really goes to the heart of what matters to us. Let us elaborate: We have been motivated to conduct this research by a “real-world” task in which we found the performance of the available computer vision methods lacking. By real world task we mean: An application of business value for a certain business case that hinges on the (classification or segmentation) accuracy of a computer vision system operating on high-resolution images taken in the real world. The problem manifested itself with rotated such images, and existing methods (e.g., equivariant architectures, data augmentation) did not scale to our problem: They increased the training and/or inference time to the point that invalidated the business case. This inspired research into solving in-plane rotation of single objects on large, real-world-ish images _efficiently_.
> >
> > Against this background, what makes a setting “real-world” for us is rather its scale and image content than if rotation was introduced synthetically.
> >
> > We added the following sentence at the beginning of the introduction to explain what we mean by “real-world” and ask the reviewer to consider our point of view:
> >
> > “[…] This is a highly desirable property for all vision systems, especially if they are to be deployed in real-world settings that are characterized by significant scale and visual complexity […]”.
> >
> > 15.	*Nr. 22: The authors have introduced a number of typos in their revision, contrasting with the quality of the original text. Examples: 1) "This demands methodological simplicity with to low development overhead and fast execution w.r.t. runtime on top of excellent performance on various downstream vision tasks (e.g. classification or segmentation)." 2) "While improving rotational stability do data agumentation approaches not tackle the issue on a fundamental level."*
> >
> > Please excuse the lacking text quality – we have done another spell-check by two authors. In addition, we will read over the final text again tomorrow before submitting the final version, if permitted by the platform.
> >
> > If there are any additional language issues you spot, please point them out to us.
> >
> > 16.	*Nr. 23: This was not noticed previously, but the authors seem to have forgotten to add an introduction \section to the paper and are recommended to do so.*
> >
> > Thank you for the catch - we added the section.

---

> > > ### Comment · Reviewer_iCG2 · 2025-01-14
> > >
> > > The authors' engagement is appreciated.
> > >
> > > 1. W.1: The characterization of the method as one that "separates the finding of the angle of rotation of a given input from subsequent rotating it back to its canonical appearance before further processing, thus performing an artificial version of mental rotation" does not uniquely describe the work and could be said equally of Boominathan et al. (2016) or Fang et al. (2020). The overlap with prior work in both the mental-rotation inspiration and the canonicalization-as-preprocessing approach should be explicitly acknowledged.
> > >
> > >    How does the proposed method differ materially from Boominathan et al. (2016)? That the authors train a model to directly estimate image rotation rather than classifying it? Highlighting again the necessity of comparison, this design decision was discussed in Boominathan et al. (2016):
> > >    > Rotation compensation of an image can be achieved in one of two ways. We can either
> > >    > - train a model to estimate rotation of a given image. Use that estimate to obtain the absolute-zero image.
> > >    > - train a model to detect whether a given image is rotated or not. Given an image, rotate that image through all possible angles. Select the image which the model believes is not rotated.
> > >
> > >    > Admittedly, the second option is more convoluted than the first. However, we still use this approach because it is simpler problem to solve. A rotation estimation model has to learn internal representations of how images at all absolute angles look like. A rotation detection model just needs to have a representation of how an absolute-zero image looks like. Given that there can be large amount of variability in how an absolute-zero image itself may look like (varying image detail and nuisance factors), the task of rotation detection itself is a challenging one to solve.
> > > 2. W.2: It remains inappropriate to claim that "AMR remains the most potent way of addressing rotations" in any context. The authors may claim that AMR outperforms a model trained with random-rotation augmentation, but "most potent" implies that AMR outperforms relevant baselines, against which it was not evaluated. The concern relates not to dataset scope, but to comparisons.
> > > 3. W.5: The authors seem to have forgotten to copy their bullet #4, as they made changes in the text. It appears the concern outlined originally was not clear. The limitation was not that "such a strict inductive bias is a trade-off since it will most likely lead to sub-optimal learning in the presence of transformations that can not be encoded within that bias," but that AMR assumes that the training dataset has been pre-canonicalized relative to planar rotation, and that the assumption does not hold for ImageNet.
> > > 4. Nr. 7: The authors are not requested to provide examples of foundation models, but to lay the foundation that, for such models, "rotation invariance is needed to solve the problem of degraded performance in the presence of in-plane rotation." It remains unclear whether models with a "modern, large architecture" trained on "correspondingly large datasets" share such a susceptibility. Given the newly refocused message of the paper, the authors should consider replacing the segmentation model in the evaluation of Section 6 with SAM (Kirillov et al., 2023). The authors also should not claim that AMR offers *equi*variance.
> > > 5. Nr. 9: The clarification should be added throughout, including in the abstract ("We test it on ImageNet, Stanford Cars, and Oxford Pet."), figures, and tables.
> > > 6. Nr. 17: The authors are requested to note the architectures/parameter counts for Worrall et al. (2017), Laptev et al. (2016), Cohen & Welling (2016), and Marcos et al. (2017). Each can refer to multiple variations.
> > > 7. Nr. 20: The paper lends the understanding that the authors train separate AMR for each base model: "We then train two AMR modules in conjunction with each upright trained base model, one for 33 epochs and the other for 5." The reader is left wondering whether increases in performance as model size increases are due to improvements in AMR or the ability of the downstream model. The authors should consider reporting rotation-classification accuracy for *each* model. In the authors' framing of AMR as an "add-on," agnostic of the downstream model, it seems as though it might be valuable to understand how large an AMR has to be to improve downstream performance. The observation that "base-networks exhibit some robustness to small rotations and the input only has to be upright 'enough'" is also relevant to the authors' motivated "practical applicability" and should be separately evaluated.
> > > 8. Nr. 21: The authors' claim that "existing methods . . . did not scale to [their] problem" has not been demonstrated. It must be pointed out that the authors backtracked on their earlier rebuttal commitment to include evaluations against relevant baselines.

---

> > > > ### Author Response · Authors · 2025-01-14
> > > >
> > > > 1. Thank you for giving us the opportunity to settle this uncertainty. Yes, the differences start with that we fully develop an approach that Boominathan et al. briefly sketch and reject (first bullet point in your qote). As we argue below (#8), the intellectual and scientific challenges in undertaking this are substantial. Boominathan et al. state that “A rotation estimation model has to learn internal representations of how images at all absolute angles look like.” and opt for solving a simpler problem (rotation detection) after the 1-sentence sketch of the approach. We instead solve rotation estimation.
> > > > To further highlight the previous contribution of Boominathan et al. we added the following clarification to the introduction:
> > > >
> > > >    “[...] thus performing an artificial version of mental rotation. The problem of rotation estimation has been considered hard in the literature before \citep{boominathan2016compensating}. However, [...]”.
> > > >
> > > >
> > > > 2. We changed the formulation to avoid any unwanted implications:
> > > >
> > > >    “[...] Our core findings are replicated on both datasets: Rotating the images reduces all the models' performances and AMR remains the more powerful way of addressing rotations on these datasets. [...]”
> > > >
> > > >
> > > > 3. Yes, this assumption does not strictly hold for ImageNet, but it holds well enough that our AMR can be trained without an issue (cp. the results). This is similar to typical assumptions in self-supervised learning (e.g., next word prediction on web-scraped texts), where it is understood that the self-supervision signal is noisy (e.g., there might be wrong words or typos in text found on the web), but not to the point of impairing training beyond acceptable levels. We made a note to this regard in the text:
> > > >
> > > >     “[...] learning in the presence of transformations that can not be encoded within that bias (e.g. rotations off-plane from the image itself). Most datasets such as ImageNet likely contain a wide variety of such perturbations. They can make the self-supervised learning signal more noisy but our results show that they do not critically impair the AMR training [...]”
> > > >
> > > > 4.  Sorry, we missed that meaning of the earlier comment. We now added an additional relevant citation that makes this point (of foundation models not generally being invariant to rotation) w.r.t. SAM (Kirillov et al., 2023):
> > > >
> > > >     “[...] These large and ever-evolving architectures are generally not rotation invariant (Mondal et al., 2023) [...]“.
> > > >
> > > >     *The authors also should not claim that AMR offers equivariance*
> > > >
> > > >      We agree and changed it to invariance.
> > > >
> > > > 5. Yes. We have extended the abstract to read:
> > > >
> > > >    “[...] We test it on randomly rotated versions of ImageNet, [...]”
> > > >
> > > >    Figures and tables already clearly state when rotated and when upright versions of the data are used.
> > > >
> > > > 6. Worrall et al. (2017) only report one architecture on rotated-MNIST (Table 1), hence the citation is fully precise (or do we miss something here?).
> > > >
> > > >     Laptev et al (2016) also only report one result on rotated-MNIST (Table 2).
> > > >
> > > >     Cohen & Welling (2016) report multiple results so we amended the text to clearly refer to the P4CNN implementation, which is their most powerful.
> > > >
> > > >     Marcos et al. (2017) have adapted versions of their base implementation they call “only scalar field” and “test-time augmentation”, therefore we indicated that we report the base implementation.
> > > >
> > > > 7. There are two points to address here:
> > > >
> > > >     (1) The size of the AMR module does not at all depend on the downstream model. Is this misconception insinuated by the following paragraph?
> > > >
> > > >     “We train all of our base CNNs for $100$ epochs on ImageNet, and the vision transformer is trained for $300$ epochs, in accordance with the training recipes for the torchvision base models.  We then train two AMR modules in conjunction with each upright trained base model, one for $33$ epochs and the other for $5$.”
> > > >
> > > >     Please let us know what gave that impression so we can improve the clarity of our text, respectively.
> > > >
> > > >     (2) The ability of the base models to deal with (fully) rotated inputs is given in the corresponding Tables (1 and 2)  - hence the improvement from rotated testing to AMR is already reported in the paper. It might be interesting to further investigate the robustness of various base architectures against small jitters in the inputs. This is not the focus of our work so we leave further studies to exactly quantify the extent to which BMs are robust to small rotations to future work.

---

> > > > > ### Author Response · Authors · 2025-01-14
> > > > >
> > > > > 8. The mentioned existing methods were never intended to work on larger problems (or: to scale), as they were originally published in a time at which respective datasets (e.g. ImageNet) would have been available for consideration. However, the original authors chose to work on MNIST-like datasets that in turn we consider not to be our “problem” (we would rather say: not our focus / domain / task).
> > > > >
> > > > >     One could argue that it has merit to port them to such tasks and see how they fare. We would agree. But take Boominathan et al. as an example (a work from 2016 with manageable impact so far): It implements a related (in their words: simpler) approach with a 3-layer CNN. Scaling it to ImageNet, in our experience, can be likened to scaling the ideas behind LeNet-5 to AlexNet. This (scaling LeNet-5) is a lot of original research that is unrelated from building AlexNet in the first place. In the literature (and staying in this example), the authors of AlexNet referenced LeNet-5 rather than using some naive scaled-up version of LeNet-5 as a baseline. Scaling “concepts” up is work that stands on its own, and is out of scope of our work here.

---

### Review · Reviewer_JMsf · 2024-12-12

**Summary Of Contributions:**

The paper concerns tasks with images with a well-defined up-direction. The paper proposes to classify the rotation of an image by using features from a frozen model trained on upright images as additional input to an angle regressing network that is trained supervised on rotated images.

**Audience:**

Yes

**Claims And Evidence:**

No

**Requested Changes:**

I will recommend acceptance of the paper is the weaknesses described above are remedied, as I believe that the criteria for TMLR acceptance are fulfilled then.

**Strengths And Weaknesses:**

### Strengths
- The approach is interesting. In particular the feedback from pretrained features to the rotation regressor is underexplored in prior work as far as I know.
- The detailed test with printed images that were re-photographed from different angles goes far beyond the evaluations that are usually seen.


### Weaknesses
- The related work section is incomplete:
    - Works on canonicalization are missing. There are a number of deep learning works on canonicalizing data wrt known data transformations [1,2,etc]. The present paper should be placed in this context.
    - GCNNs have been generalized to steerable CNNs, which remove the linear scaling with the number of rotations [3,4].
- The evaluations are incomplete without reporting number of parameters and runtime/flops for different approaches.

[1] Kaba, Sékou-Oumar, et al. "Equivariance with learned canonicalization functions." ICML 2023.

[2] Mondal, Arnab Kumar, et al. "Equivariant adaptation of large pretrained models." NeurIPS 2023.

[3] Cohen, Taco S., and Max Welling. "Steerable cnns." ICLR 2016

[4] Weiler, Maurice, and Gabriele Cesa. "General E(2)-equivariant steerable cnns." NeurIPS 2019.

---

> ### Author Response · Authors · 2024-12-16
> **Thanks for review and intended changes to manuscript**
>
> Many thanks for the review. We agree on the weaknesses and are currently working on incorporating your suggestions into the paper.

---

> ### Author Response · Authors · 2025-01-12
>
> We have re-shaped the paper to include references and a discussion of missing related works with a focus on canonicalization functions, including revised novelty claims. We have sharpened the focus of our contribution towards the practical viability of our approach we achieve through the disentanglement of our method with the base architecture instead of SoTA comparisons on small datasets (i.e. MNIST).
> We added the number of added parameters and flops within the text discussion to not overload the tables.

---

> > ### Author Response · Authors · 2025-01-21
> >
> > Dear Reviewer
> > We have iterated on the paper considerably since the initial version and we are satisfied with the state it has now reached. To facilitate the comparison between the revised and initial versions we have compiled a log of the important changes.
> >
> > 1.	Re-focus main claim of paper towards disentanglement of rotation invariance and classification. Adapted claims of novelty to properly reflect the existing work.
> > 2.	Discussion of previously missed related work.
> > 3.	Added a small section showing how our method can be framed as a special case of the canonicalization functions framework (Kaba, Sékou-Oumar, et al. ICML 2023.) and contrasting our work with the existing methods.
> > 4.	Description of the fixed overhead (in terms of FLOPS and parameter) introduced by the AMR module.
> > 5.	Changes to Figure 1. To better illustrate our claim that (naïve) data augmentation reduces the sample efficiency of the training process.
> > 6.	Added a section in the appendix investigating the angle-classification performances of the AMR modules.
> > 7.	Various textual and clarity improvements.

---

### Review · Reviewer_UuGr · 2024-12-29

**Summary Of Contributions:**

This paper presents artificial mental rotation (AMR), a method for adding rotational invariance to CNNs and ViTs. Unlike most related methods, which usually fall into one of two broad categories based on invariant NN architectures or data-augmentation, AMR creates canonical 'un-rotated' versions of the input images and can be seen as a data-preprocessing step. AMR compares favoribly with data-aumgentation on large-scale tasks such as ImageNet, as well as against invariant NN architectures for smaller datasets such as MNIST. The paper includes several ablation studies which support the method and results.

**Audience:**

Yes

**Claims And Evidence:**

No

**Requested Changes:**

The following changes would be required in order to secure my recommendation:
1. Rewrite the paper in the context of the exisitng literature on cannonicalisation, including appropriate references and reducing the novlelty claims of the paper. For example, I would make the following changes (among others):
    * In the abstract, do not claim that AMR is a "novel deep learning paradigm" but rather a spcific instance of the exisitng canonicalisation paradigm.
    * In the introduction and related work sections, include references and disucssions of the papers mentioned in the above section. E.g.:
        * The second last paragraph of the introduction should no longer imply that such a 'third way' for solving this problem had not previously existed.
        * Contribution (a) in the last paragraph of the intro should be removed.
        * Similarly contribution (d) shoud be removed since many state-of-the-art cannonicalistion methods have not been compared against.
2. Add experiments comparing against some of the canonicalisation methods mentioned in the previous section.
3. Add discussion of learned / instance-specific data autmentation methods.

The following changes would not be required, but would strengthen the work:

a) Add experements to compare against some learned and instance-specific data autmentation methods.

**Strengths And Weaknesses:**

## Strengths

* The paper is clearly written and easy to understand.
* The method works well in several settings.
* The authors provide interesting and useful ablations studies.

## Weaknesses

* The paper positions AMR–and more braodly the idea of creating a canoonical version of an input image which is then fed into a classifier–as a novel method for solving the problem of invariant classification. However, there is already an well-developed literature on this approach. A non-exhaustive list of cannonicalisation methods includes: Allingham et al. (2024), Kaba et al. (2023), Mondal et al. (2023), and Kim et al. (2023). Please see related work sections of these papers for additional key references.
* Similarly, the authors claim that they compare against state-of-the-art methods. However, they only compare with simple data-augmentation. There is a wealth of methods for learned and/or instance-specific data-augmentation methods which have not been mentioned or compared with. Examples include Miao et al. (2023) and Immer et al. (2022). Please see related work sections of these papers for additional key references.

## References

James Urquhart Allingham, Bruno Kacper Mlodozeniec, Shreyas Padhy, Javier Antorán, David Krueger, Richard E. Turner, Eric T. Nalisnick, José Miguel Hernández-Lobato:
A Generative Model of Symmetry Transformations. CoRR abs/2403.01946 (2024)

Sékou-Oumar Kaba, Arnab Kumar Mondal, Yan Zhang, Yoshua Bengio, Siamak Ravanbakhsh:
Equivariance with Learned Canonicalization Functions. ICML 2023: 15546-15566

Arnab Kumar Mondal, Siba Smarak Panigrahi, Oumar Kaba, Sai Mudumba, Siamak Ravanbakhsh:
Equivariant Adaptation of Large Pretrained Models. NeurIPS 2023

Jinwoo Kim, Dat Nguyen, Ayhan Suleymanzade, Hyeokjun An, Seunghoon Hong:
Learning Probabilistic Symmetrization for Architecture Agnostic Equivariance. NeurIPS 2023

Ning Miao, Tom Rainforth, Emile Mathieu, Yann Dubois, Yee Whye Teh, Adam Foster, Hyunjik Kim:
Learning Instance-Specific Augmentations by Capturing Local Invariances. ICML 2023: 24720-24736

Alexander Immer, Tycho F. A. van der Ouderaa, Gunnar Rätsch, Vincent Fortuin, Mark van der Wilk:
Invariance Learning in Deep Neural Networks with Differentiable Laplace Approximations. NeurIPS 2022

---

> ### Author Response · Authors · 2025-01-01
> **Reshaping the scope of the Paper**
>
> Many thanks for the informative review. Apparently, we had a significant blind spot regarding existing work on canonicalization. We are taking the remaining time to reshape the intro/related work sections and add relevant baselines.

---

> ### Author Response · Authors · 2025-01-12
>
> We have re-shaped the paper to include references and a discussion of missing related works with a focus on canonicalization functions, including revised novelty claims. We have sharpened the focus of our contribution towards the practical viability of our approach we achieve through the disentanglement of our method with the base architecture instead of SoTA comparisons on small datasets (i.e. MNIST).

---

> ### Author Response · Authors · 2025-01-21
>
> Dear Reviewer
> We have iterated on the paper considerably since the initial version and we are satisfied with the state it has now reached. To facilitate the comparison between the revised and initial versions we have compiled a log of the important changes.
>
> 1.	Re-focus main claim of paper towards disentanglement of rotation invariance and classification. Adapted claims of novelty to properly reflect the existing work.
>
> 2.	Discussion of previously missed related work.
> 3.	Added a small section showing how our method can be framed as a special case of the canonicalization functions framework (Kaba, Sékou-Oumar, et al. ICML 2023.) and contrasting our work with the existing methods.
> 4.	Description of the fixed overhead (in terms of FLOPS and parameter) introduced by the AMR module.
> 5.	Changes to Figure 1. To better illustrate our claim that (naïve) data augmentation reduces the sample efficiency of the training process.
> 6.	Added a section in the appendix investigating the angle-classification performances of the AMR modules.
> 7.	Various textual and clarity improvements.

---

### Comment · Reviewer_JMsf · 2025-01-31
**Delay**

Dear all,

There is a very long discussion between reviewer iCG2 and the authors, and the paper has been substantially altered. I will need until next week to provide a final recommendation.

---

### Decision · Action_Editor_WX4t · 2025-02-11

**Recommendation:** Reject

**Comment:**

After careful consideration of the reviewers' feedback and the subsequent revisions, we regret to inform you that the paper cannot be accepted in its current form.

The reviewers consistently noted that it does not sufficiently engage with prior work in this area. Despite the addition of Section 7, the discussion on related literature remains inadequate, and key comparisons—both conceptual and experimental—are missing. The reviewers also pointed out that while some surface-level revisions were made, the more substantial improvements necessary to position the work within the broader research landscape were not fully addressed.

A significant concern raised was that the paper’s claims have been adjusted to the point where its relevance to the intended audience is unclear. Moreover, the method's lack of rotation invariance, as evident in Figure 5, and its susceptibility to adversarial perturbations further weaken its impact. Reviewers also noted that the rebuttal commitments were not fully realized in the revisions, which limits confidence in the current state of the work.

Reviewers indicated that the authors’ revisions have led to a situation where the claims have been scaled down so much that the paper's relevance to the intended audience is questionable. This suggests that the work does not present strong, well-supported contributions in its current form.

While the topic remains of interest, the manuscript requires substantial revisions, particularly in providing thorough experimental comparisons, refining its claims to align with its contributions, and better contextualizing its novelty in light of existing approaches. We encourage the authors to undertake these revisions and consider resubmitting a significantly improved version in the future.

**Audience:**

Reviewers indicated that the authors’ revisions have led to a situation where the claims have been scaled down so much that the paper's relevance to the intended audience is questionable.

**Claims And Evidence:**

Based on the reviewers' feedback, the claims made in the submission are not sufficiently supported by clear and convincing evidence. Several concerns were raised:

The discussion added in Section 7 does not sufficiently differentiate the proposed approach from existing methods. Reviewers pointed out that prior work (e.g., Mondal et al.) also predicts and applies a rotation angle in a similar manner, raising concerns that the paper lacks adequate evidence to substantiate its claims of novelty.

Despite requests for substantial changes, the revisions primarily consisted of surface-level modifications rather than adding necessary experimental evaluations. The absence of comparisons to relevant baselines weakens the evidence supporting the paper’s contributions.

The reviewers pointed out that the proposed method is not truly rotation-invariant, as shown in Figure 5. This directly undermines one of the core claims of the paper. Additionally, the susceptibility of the method to adversarial perturbations, despite being motivated by robustness concerns, further diminishes its impact.

Overall, the reviewers’ comments indicate that the submission lacks the necessary depth in experimentation, justification, and discussion of prior work to convincingly support its claims.

**Resubmission Of Major Revision:**

The authors may consider submitting a major revision at a later time.